# Highly twisted supercoils for superelastic multi-functional fibres

Wonkyeong Son[1], Sungwoo Chun[2], Jae Myeong Lee[1], Yourack Lee[3], Jeongmin Park[3], Dongseok Suh [3], Duck Weon Lee[4], Hachul Jung[5], Young-Jin Kim[5], Younghoon Kim[6], Soon Moon Jeong[1], Sang Kyoo Lim[1] & Changsoon Choi [1]

Highly deformable and electrically conductive fibres with multiple functionalities may be useful for diverse applications. Here we report on a supercoil structure (i.e. coiling of a coil) of fibres fabricated by inserting a giant twist into spandex-core fibres wrapped in a carbon nanotube sheath. The resulting supercoiled fibres show a highly ordered and compact structure along the fibre direction, which can sustain up to 1,500% elastic deformation. The supercoiled fibre exhibits an increase in resistance of 4.2% for stretching of 1,000% when overcoated by a passivation layer. Moreover, by incorporating pseudocapacitive-active materials, we demonstrate the existence of superelastic supercapacitors with high linear and areal capacitance values of 21.7 mF cm$^{-1}$ and 92.1 mF cm$^{-2}$, respectively, that can be reversibly stretched by 1,000% without significant capacitance loss. The supercoiled fibre can also function as an electrothermal artificial muscle, contracting 4.2% (percentage of loaded fibre length) when 0.45 V mm$^{-1}$ is applied.

[1] Division of Smart Textile Convergence Research, DGIST, Daegu 42988, South Korea. [2] Department SKKU Advanced Institute of Nanotechnology (SAINT), Sungkyunkwan University, Suwon, Gyeonggi-do 16419, South Korea. [3] Department of Energy Science, Sungkyunkwan University, Suwon, Gyeonggi-do 16419, South Korea. [4] Center for Self-powered Actuation, Department of Biomedical Engineering, Hanyang University, Seoul 04763, South Korea. [5] Medical Device Development Center, Osong Medical Innovation Foundation, Cheongju, Chungbuk 28160, South Korea. [6] Convergence Research Center for Solar Energy, DGIST, Daegu 42988, South Korea. These authors contributed equally: Wonkyeong Son, Sungwoo Chun. Correspondence and requests for materials should be addressed to C.C. (email: cschoi@dgist.ac.kr)

Yarn or fibre-based one-dimensional (1-D) conductors can serve as effective and universal substrates for various wearable applications, such as textile electronics, implantable devices, and microscale devices. In addition to providing various functionalities, these fibres need to have elastic properties so that the devices can be highly deformable without performance loss during application of tensile strain, which is important for more advanced and human-friendly applications[1–3].

Recently, important advances have been made toward realising highly stretchable one-dimensional (1-D) conductors. To achieve both high conductivity and stretchability, fibre conductors have been developed by either incorporating conducting additive materials into elastic substrates or designing microscopically or macroscopically stretchable structures. From a material aspect, composite fibres consisting of conductive metal fillers and elastomeric substrates have been introduced widely. For example, Ag nanoparticles or nanowires were embedded within elastomer fibres using various spinning technologies[4–6]. These metal/polymer composite fibres exhibit excellent length-normalised conductivity and high stretchability; however, they still suffer from significantly increased resistance upon a threshold strain. Alternatively, a liquid metal such as a GaIn was injected in hollow elastomer fibres[7,8]. Although the fibres injected with a liquid metal were reported to have excellent stretchy-invariant electrical conductivity because of high viscosity of the liquid metal, this method is based on expensive elements and needs to be contained in hollow structured containers to prevent leakage, which limits wider application. On the other hand, from a structural aspect, microscopically buckled structures[1,9–11], macroscopically coiled structures[2,3,12–16], or a combination of both[17] were employed for carbon nanotube (CNT) yarns or twist-spun CNT sheet-wrapped fibres to provide structural stretchability. These buckled or coiled fibres exhibited relatively high electrical and elastic properties and were demonstrated to be effective electrodes for elastic sensors[1,10,11] and energy storage[2,3,9,10,15].

Ultimately, however, realising highly elastically deformable fibrous conductors with microscale diameters and assigning them with multiple functionalities remain challenging. Here, we report on achieving both high stretchability and highly improved quality factor (percent strain divided by percent resistance change). The supercoil structure of DNA is mimicked in a twisting and bending of a double helical axis that not only affects structural transitions and interactions[18], but also enables tight compaction to store substantial genetic information within the cell. Although the coiling strategy has been previously reported, the presented supercoil structure has not been reported, to the best of our knowledge. The presented supercoiled fibres are fabricated by highly over-twisted insertion. Due to the high degree of structural compaction, by storing large linear strain in the fibre tensile direction, the supercoiled fibres exhibit superelasticity (~ 1500%) without significant electrical fracture. This high degree of structural organisation can store the ultra-high linear strain, and thus allow the supercoil structure to be used for extendable conductive transmission lines, pseudocapacitive supercapacitors, and artificial muscles.

## Results

**Fabrication and morphology of supercoil fibre electrodes.** The schematic illustrations of the supercoiled fibres and buckled surface are presented in Fig. 1a and its inset, respectively. We used spandex fibres wrapped a in CNT-spun sheet (spandex@CNT fibre) as starting materials. Such spandex fibres (~ 200 μm diameter) are stretchable, low-cost, commercially

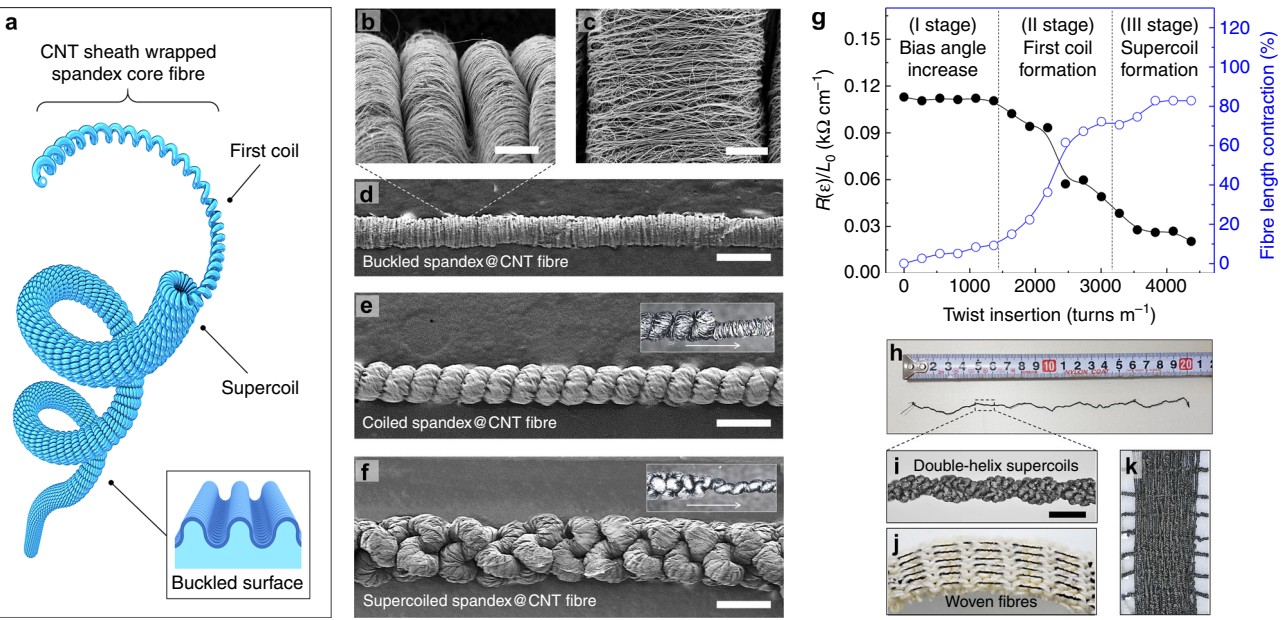

**Fig. 1** Schematic illustration and morphological information of supercoiled fibres. **a** Schematic illustration of highly twisted spandex@carbon nanotube (CNT) fibre, consisting of first-coils and supercoils. Inset shows a schematic of microscale buckled surface. Scanning electron microscopy (SEM) images for **b** microscopic scaled buckles formed on the surface of spandex@CNT fibre relaxed from pre-strain (scale bar = 15 μm), and **c** its magnification showing uniaxially aligned CNT bundles (scale bar = 5 μm). SEM images for **d** noncoiled, **e** first coiled, and **f** supercoiled spandex@CNT fibres, showing subsequent structure transformations as the number of twists increases (scale bars for **d**, **e**, and **f** are 200 μm). Inset images from **e** and **f** show fibre transformations from noncoiled fibre into first coil, and from first coil into supercoil, respectively. The arrows indicate the propagation direction of coil formation. **g** Measured length-normalised resistance changes and fibre length contraction versus number of inserted twists until full supercoiling ($R(\varepsilon)$ is the resistance of fibre at the stretched state and $L_0$ is the length of fibre at the initial state). Photograph of **h** free-standing state, double-helix structured, supercoiled fibre, and optical images of **i** its magnification (scale bar = 1 mm), **j** 3.5-cm-long, six-woven supercoiled spandex@CNT fibres into a commercial mock rib-structured textile, and **k** 20 mm-long, 7 mm-wide supercoil textile consisting of 27 spandex@CNT fibres

available in effectively unlimited length, and require no treatment before use. More specifically, the spandex fibre core is fully stretched in the tensile direction (fabrication strain, $\varepsilon_{fab} = 400\%$) and conducting CNT sheath layers mechanically drawn from multiwalled forests[19] were wrapped on the stretched fibre using a custom-made twisting machine (Supplementary Fig. 1). The CNT orientation was parallel to the core fibre length direction during wrapping. Ethanol was subsequently dropped on the fibre for surface tension-based densification of the CNT sheets during ethanol evaporation. The length of the relaxed spandex@CNT fibre was longer than the pristine spandex because of the constraint by the mechanically robust CNT sheath wrapping[1]. Relaxation of pre-stretched fibre after CNT wrapping resulted in a buckled microstructure on the fibre surface, as shown in scanning electron microscopy (SEM) image presented in Fig. 1b, consisting of uniaxially aligned CNT bundles in the fibre length direction (Fig. 1c).

The micro-buckled fibres relaxed from fabrication strain (Fig. 1d) became highly deformable elastically by inserting a sufficiently high twist for supercoiling. One end of the relaxed spandex@CNT fibre was attached to the electrical motor for the insertion of the twist while the other was fixed. The supercoiling process can be categorised into three stages. The first stage is a CNT sheath bias angle increase. In the initial twisting stage, only the CNT sheet bias angle ($\theta$, the angle between the CNT bundle alignment direction and the fibre axis) increases up to 48.3° just before the first coil formation (Supplementary Fig. 2), showing no observable macroscopic changes in fibre morphology. In the second stage, after sufficient application of twisting, the microscopically buckled spandex@CNT fibre is fully transformed into a macroscopically coiled fibre, as shown in the SEM images in Fig. 1e. This first coil formation starts at a twisting of about 1350 turns m$^{-1}$ (relative to the initial fibre length just before twisting insertion), and the coiling process propagates along the fibre length direction (inset of Fig. 1e). The last stage is a supercoil transformation in which the coiled fibres undergo coiling once again, resulting in a highly packed structure, when more than 3000 turns m$^{-1}$ twisting is inserted (Fig. 1f). The fibre transformation from the first coil into a supercoil image is shown in the inset of Fig. 1f. The surface tension-based densification-induced binding of the CNT sheets onto the spandex fibre was so strong that no noticeable delamination was observed during the giant twist insertion until supercoiling.

Structural differences of the supercoil with other coils are presented in the schematic illustrations (Supplementary Fig. 3). For examples, first coil fibres (Supplementary Fig. 3b) are widely reported for various stretchable fibrous applications such as supercapcaitors[2,3,15,17], actuators[20,21], and high toughness[22], and can be fabricated by moderate twisting insertions. Due to the simple structure, however, first coil fibres have normally reported less than 400% stretchability. Helical-coil fibres are different from other coils in that they are fabricated by not just twisting but by winding active fibres onto the surface of non-active core substrate fibres (Supplementary Fig. 3c)[23–25]. Such helical coil fibres are reported to be stretchable up to 800% when the core substrates have appropriate elasticity[25]. However, they can suffer from low specific performances when normalised by system dimension including thick and bulk core substrate. This is because the as-used core substrate does not contribute to the fibre functionalities such as mechanical actuation or energy storage, but just provides a mechanical stretchability, which might especially lead to low volumetric or surface areal capacitance, energy, or power densities.

The electrical and morphological properties of the fibre before fully supercoiling were dynamically changed during the twisting as shown in Fig. 1g. The initial electrical resistance normalised by the maximum (fully stretched) fibre length of the noncoiled spandex@CNT fibre significantly decreased by 82% when fibres

were fully supercoiled; this occurred because the electrical pathway along the fibre length was enlarged by the intercoil contact area increase after the supercoil formation. The relaxed fibre length from the fabrication strain continuously decreased during twist insertion; about 67% and 83% of the fibre length contracted after the first and supercoil formations, respectively. By contrast, the total diameter for the first coiled and supercoiled fibres discretely increased by 33.6% and 113% over the noncoiled fibre, respectively, roughly showing the volume conservation of the total fibre. Due to the ultra-high twist insertion, the supercoiled fibre is untwisted when both ends of the fibre are not tethered. Torque-balance locking can be an effective way to prevent this unwanted untwisting. Photograph of the double-helix structured supercoiled fibre made by self-inter locking is shown in Fig. 1h. This type of internally torque-balanced structure eliminates the need for external torsional tethering. Plying a single, fully supercoiled fibre in the direction opposite to the internal twist of the fibre creates a structure in which the chiralities of fibre twist and plying are opposite by double-helix structure (Fig. 1i)[26]. This was accomplished by folding the supercoiled fibre itself, while prohibiting relative rotation of fibre ends. Another possible strategy to prohibit untwisting is to upscale the supercoil fibres into a neat textile structure because the supercoiled fibres could be mechanically tethered by adjacent fibres. The supercoiled fibres were mechanically strong enough so they can be woven into a commercial mock rib-structured textile (Fig. 1j) or be assembled into the textile by themselves (Fig. 1k). In addition, the over-twisting process is stable for scalable supercoil fabrication. Hence, much longer supercoil fibre (60 cm in length), fabricated using a 3-m long spandex fibre, with uniform morphological and electrical properties was successfully demonstrated, showing resistance-linearity property with 100 Ω cm$^{-1}$ slope value (Supplementary Figs. 4 and 5).

**Structure analysis of resulting supercoiled fibres.** Our unique structures that contribute to high fibre stretchability can be categorised into two parts: one is microscopic scale buckles, and the other is macroscopic scale supercoils. The morphological width of CNT micro-buckles is investigated against pre-strain (Fig. 2a) applied to a bare, noncoiled spandex fibre before CNT wrapping, and CNT loading layers (Fig. 2b). Upon observations, the width of the buckles is roughly inversely proportional to the degree of pre-strain application (CNT loading level is fixed as five layers in this case). The average width of the CNT buckles formed from relaxation of 100% pre-strained fibre was about 38 μm and it significantly decreased to 16.3 μm by 400% pre-strain relaxation. The effect of CNT loading level on buckles formation, where pre-strain is fixed at 400%, showed slight increase of buckle width from 14.6 to 20 μm as the number of CNT wrapping layer increased from 1 to 9. The fibre contraction force to recover initial length after pre-strain relaxation is inversely proportional to the fibre length ratio of before to after relaxation ($\Delta L/L_0$) as shown in Fig. 2b. The larger contraction force by either larger pre-strain application or lower CNT loading level results in narrower buckle width. Meanwhile, a main experimental parameter to control various morphological conditions of the supercoil (Fig. 2c) is the total twisting number inserted to complete the full supercoil formation, where $D$ is a diameter of first coiled fibre, $d$ is a diameter of noncoiled spandex@CNT fibre, $\theta'$ is a coil bias angle, and $D'$ and $L$ are diameter and length of the supercoiled fibre, respectively. Linear coil density (number of coils per unit length) and supercoil index ($D'/d$) are plotted versus twisting number insertion as shown in Fig. 2d. Here, linear coil density can be defined as $N/L$, where $N$ is total number of coils for a given fibre length ($L$), and can be calculated by dividing a fibre length

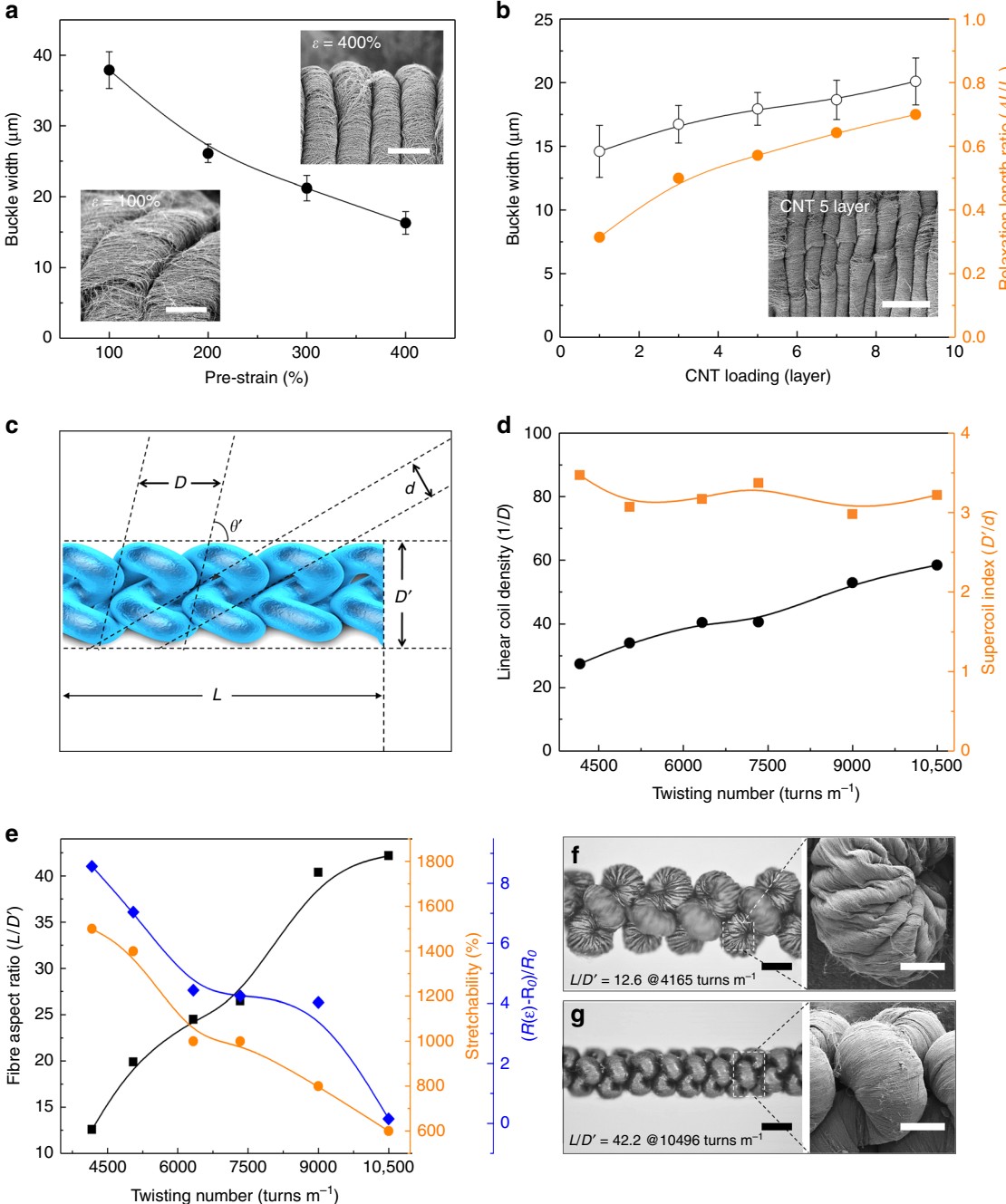

**Fig. 2** Structural analysis of microscopic buckles and macroscopic supercoils. **a** Average width of microscopic carbon nanotube (CNT) buckles of noncoiled, relaxed spandex@CNT fibre versus applied pre-strain. Insets show scanning electron microscopy (SEM) images for CNT buckles formed from 100% (lower image) and 400% (upper image) pre-strains (scale bar = 20 μm). **b** Average buckle width and fibre contraction length ratio versus CNT loading layer. Inset shows SEM image of buckles formed from five layers CNT wrapping (scale bar = 50 μm). **c** Scheme showing various morphological parameters of supercoil fibre ($D$ is a diameter of coiled fibre, $d$ is a diameter of spandex@CNT fibre, $θ'$ is a coil bias angle, $D'$ is a diameter of supercoiled fibre, and $L$ is a length of supercoiled fibre). **d** Linear coil density ($1/D$) and supercoil index ($D'/d$) versus total twisting number insertion for fully supercoiling. **e** Fibre aspect ratio ($L/D'$), maximum stretchability, and resistance change during fibre maximum stretching versus total twisting number insertion for fully supercoiling ($R(ε)$ and $R_0$ is the resistance of fibre at stretched and initial state, respectively). Optical images showing supercoil fibres fabricated from **f** low twisting (4165 turns m⁻¹), and **g** high twisting (10496 turns m⁻¹) insertions (both scale bars = 300 μm). Insets show SEM images of magnified supercoil surfaces (both scale bars = 100 μm)

by coil width ($L/D$). Therefore, the linear coil density per centimetre fibre length is $N/L = (L/D)/L = 1/D$. While the coil density linearly increased as the twisting number increases, the coil index did not show any observable or meaningful tendencies. In addition, the aspect ratio ($L/D'$) of the supercoil fibres, in case of supercoil fabrication starting from 7-cm long initial spandex fibre,

varied from 12.6 to 42.2 with increasing twisting number as shown Fig. 2e. In other words, higher twisting number causes tougher squeezing of the fibre during supercoiling, leading to thinner and longer supercoiled fibres. In this reason, both fibre stretchability and change in resistance during the fibre stretching decreased accordingly by higher number of twisting insertion as

shown Fig. 2e. The maximum stretchability of the supercoil fibre fabricated with 4165 turns $m^{-1}$ is about 1500% and steadily drops to 600% for the fibre fabricated with 10496 turns $m^{-1}$. Optical and SEM images for supercoils with low (4165 turns $m^{-1}$, Fig. 2f) and high (10496 turns $m^{-1}$, Fig. 2g) total twisting numbers were then compared. The microscopic surface from the higher twisting number inserted supercoil fibre was less buckled and largely tightened, which might be caused by the additional tensile strain generated by inserting more twists.

**The strain dependence of electrical and mechanical properties for supercoiled fibres.** As there is normal trade-off relation between the CNT sheet loading level and fibre stretchability, as previously reported[1,17], we used five-layer CNT sheets for conductive wrapping, enabling the realisation of highly stretchable and electrically conductive supercoiled fibres. A comparison of the resistance normalised by the maximum fibre length ($R(\varepsilon)/L_{max}$) versus tensile strain for supercoiled, coiled, and noncoiled spandex@CNT fibres is shown in Fig. 3a. The initial resistance of the noncoiled spandex@CNT fibre increased by about 15% with 200% strain application. For a coiled fibre, at a strain of 620%, the resistance increased up to 90% over the initial resistance. The decrease in initial resistance after coiling is caused by the increase in contact area between coils, providing a larger electrical pathway in the longitudinal direction of the fibre. The trend is more evident for the supercoiled fibre that the initial resistance drastically increased at the low strain range (below 450%), which mainly corresponds to the progressive opening of the first coils, and the supercoils during strain application. The separation between coils during supercoil fibre stretching is shown in the optical images of Fig. 3b. In the higher strain region (450%–1300%), micro-buckles unfolding rather than coil opening mainly contributed to fibre stretchability, showing very little change in resistance.

Although the maximum stretchability before electrical failure for supercoiled spandex@CNT fibres was 1500%, we used the stable and mechanically or electrically reversible strain region (1000%) to investigate the performance of the supercoiled fibre. The resistance change ratio ($(R(\varepsilon)-R_0)/R_0$), where $R_0$ is the resistance at the initial fibre length versus strain for the supercoiled fibre, was measured during 1000% stretching and releasing (Fig. 3c). The resistance recovered to the initial value after repeated tensile deformations at 1000%. The reversible resistance change against the cyclically applied maximum reversible strain ($\varepsilon_{max} = 1000\%$) is presented in the inset of Fig. 3c. The supercoil fibre exhibited stable electrical properties, with 4.06 average resistance change ratio ($R(\varepsilon)-R_0)/R_0$) with 0.12 standard deviation during repeated 1000% strain applications. Mechanical properties of the supercoil fibre is characterised by stress–strain curves (Fig. 3d). The ultimate tensile strain before mechanical fracture of the presented supercoiled fibre was about 1500%. Moreover, the mechanical properties were largely retained after repeated stretching that no significant deformation was detected after 100 times of 1000% strain loading-unloading cycles (inset of Fig. 3d). The photographs in Fig. 3e show a supercoiled fibre mounted on Vernier calipers (initial fibre length, $L_0 = 1.5$ cm) progressively stretched to $\varepsilon_{max} = 1000\%$ ($L_{max} = 16.5$ cm) and recovered to the initial state ($\varepsilon = 0\%$), respectively, without showing any observable residual deformation (the left SEM image in the inset shows a pristine supercoil and the right SEM image shows a relaxed supercoil after 1000% strain application).

**Passivation for high quality factor and transmission line applications.** Although the large and reversible resistance change of the supercoiled fibre by varying the intercoil distance is advantageous for strain sensor applications, strain-insensitive

properties are also required for use in other applications[27]. To avoid abrupt resistance increase during fibre stretching, particularly during coil opening, the supercoiled fibres were overcoated by styrene-ethylene-butylene-styrene (SEBS) layers (~ 3.85 μm thickness) using the spray coating method. This SEBS passivation layer largely prevents supercoils from directly contacting during the fibre deformation without degrading the mechanical elasticity of the supercoiled fibre; therefore, the resistance change could be dramatically reduced. The resistance increase of SEBS/supercoiled fibre during 1000% strain application was 4.2% (Fig. 4a), which is about two orders of magnitude lower than the resistance change of nonpassivated supercoiled fibres. The scheme presented in the inset in Fig. 4a schematically depicts that the SEBS passivation (green layer) is uniformly overcoated on the buckled surface of the supercoiled fibres. SEM images of the uniform SEBS coating layers on the supercoiled fibres are presented in Supplementary Fig. 6. To investigate the usefulness of a supercoil fibre with a high quality factor, the SEBS/supercoiled fibre was used as an electrical signal transmission line. A square wave signal with 20 Hz frequency generated by a function generator was transmitted to an oscilloscope through bare and SEBS-overcoated supercoiled fibres, and the resulting transmitted signal amplitudes were compared (Fig. 4b). The quality factor of the SEBS/supercoiled fibre was compared with previously reported fibre-based conductors with diameters similar to the present work (Fig. 4c). Specifically, the SEBS/supercoiled fibre showed an impressively higher quality factor ($Q = 238.8$) than (a) the buckled CNT@sandwich structured rubber fibre ($Q = 54$)[28], (b) buckled CNT@SEBS fibre ($Q = 65.1$)[10], (c) CNT@coiled nylon fibre ($Q = 0.1$)[3], (d) buckled CNT@coiled rubber fibre ($Q = 1.8$)[17], and (e) CNT/elastomer polymer fibre ($Q = 1.6$)[29]. A detailed comparison of quality factors of the present supercoiled fibre and prior-art yarn or fibre electrodes is presented in Supplementary Table 1. Elastomeric conductors with very-low-quality factors are useful as strain sensors, but other applications, such as signal transmission lines or conventional conductors, would benefit from very-high-quality factors[1]. The amplitude transmitted through the supercoiled fibre without passivation was drastically reduced when the fibre was fully stretched. However, the amplitude transmitted through the SEBS-passivated supercoiled fibre was largely retained even at $\varepsilon_{max}$ (1000%) application. This voltage amplitude retention performance was plotted versus strain for supercoiled fibres (inset of Fig. 4c). The signal from the SEBS/supercoiled fibre showed 92% amplitude retention, whereas the signal from the bare supercoiled fibre showed only 28% amplitude retention when both fibres were fully stretched.

A few groups have reported stretchable signal transmission lines with strain-insensitive resistance, which remains nearly constant even when the lines are subjected to large strains. Such properties are desired for transmitting analogue voltage or current signals, for example, which may contain vital information from sensors attached to or implanted in a human body. Combined with high signal-to-noise ratios (SNRs), these stretchable conductors are likely to be promising in developing health care and wound-monitoring systems, or other medical applications[30,31]. An effective demonstration of such resistance-invariant, highly expandable electrical signal transmission lines is electrocardiogram (ECG) measurement (Fig. 4d). ECG signals measured by the "three leads ECG" method were transmitted by a supercoiled fibre; a commercial cable was also used for comparison. Moreover, the supercoils were also applicable as highly extendable cables to transmit the audio and video signals. Figure 4e, f show photograph images for audio and video signals measurement setups, respectively. The ECG patterns, peak magnitude, and normal sinus rhythm transmitted by the SEBS/supercoiled fibre were clear, independent of strain, and highly

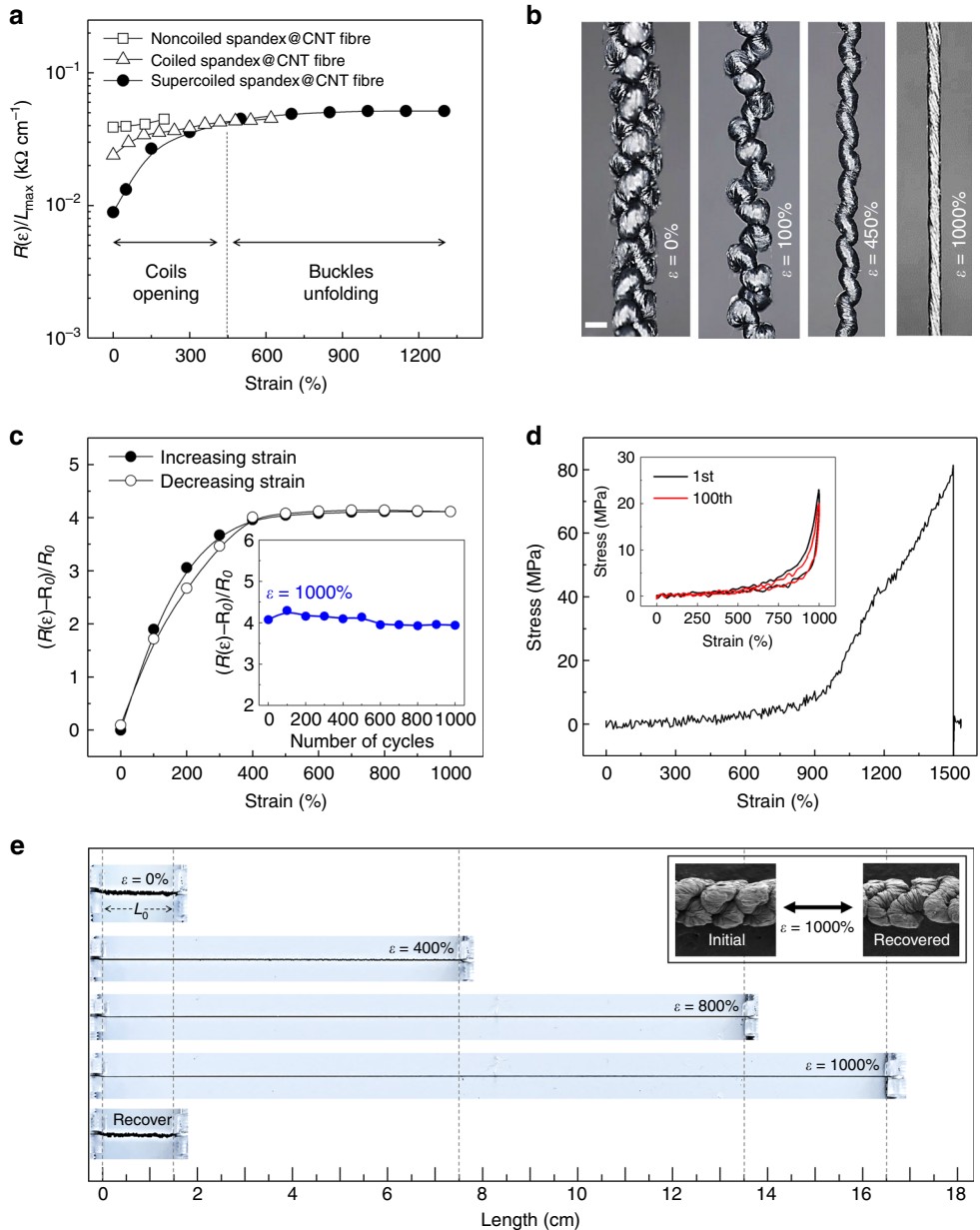

**Fig. 3** The strain dependence of electrical and mechanical properties. **a** Length-normalised resistance versus strain for noncoiled (open squares), coiled (open triangles), and supercoiled fibres (black circles). $R(\varepsilon)$ is the resistance of fibre at the stretched state and $L_{max}$ is the length of fibre at the fully stretched state. **b** Optical images for supercoil fibre showing progressive coil opening during application of 0, 100, 450, and 1000% strain (scale bars = 200 μm). **c** Resistance change ratio for a supercoil fibre versus strain for a 1000% strain loading-unloading ($R_0$ is the resistance of fibre at the initial state). Inset shows resistance change versus cycle number of 1000% strain applications. **d** Engineering stress–strain curve for 150-μm-diameter supercoil fibre. Inset shows stress–strain curves compared before (black line) and after (red line) 100th for 1000% strain loading-unloading application. **e** Photographs showing the supercoiled fibre mounted on digital Vernier calipers (the initial supercoiled fibre length was 1.5 cm). The fibre is subsequently stretched to strains of 400, 800, 1000% (16.5 cm), and recovered to the initial state. Scanning electron microscopy (SEM) images presented in the inset show (left) pristine macroscopic scale supercoil structures and (right) their relaxation after 1000% strain application

comparable to the peak from reference measurements using a commercially available cable (Fig. 4g), implying high versatility of the present passivated supercoiled fibre as an expandable cable for biomedical signal transmissions. Audio and video signals were split into the supercoil fibre and reference line simultaneously for comparison. In general, relevant frequencies range from 20 Hz to 20 kHz for acoustic electronics devices and are on the order of a few MHz for composite video signals; these frequencies are much higher than those considered in ECG measurements[27]. Real-time signal transmissions using passivated supercoil fibres under dynamic strain and comparisons with reference signals were

shown for audio (Fig. 4h) and video (Fig. 4i) signals. The transmission of video signals requires much higher accuracy in terms of the signal period (or frequency) and amplitude because the information on brightness, colour, and synchronisation of a signal is delivered at the same time[27]. Specifically, various significant signals such as horizontal synchronisation, colour reference burst, active pixel region, and front and back porch should be transmitted without delay and distortion. The transmitted audio and video signals by supercoil fibres were almost identical with that of reference cables without distorting the original information in terms of its resistance and frequency.

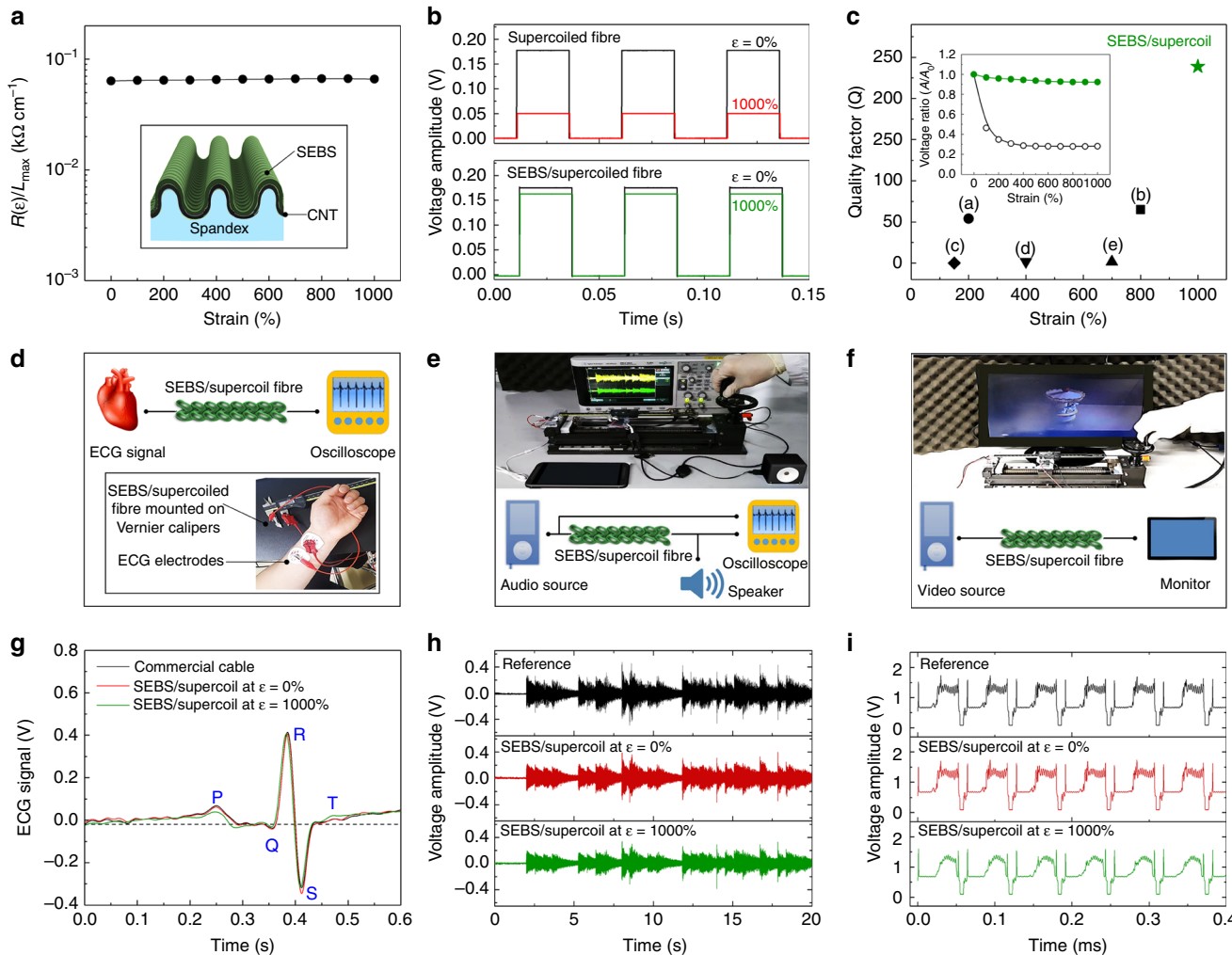

**Fig. 4** Passivated supercoiled fibre for high quality factor and transmission line applications. **a** Length-normalised resistance versus strain for styrene-ethylene-butylene-styrene (SEBS)-overcoated, spandex@carbon nanotube (CNT) supercoiled fibre. Inset shows the scheme for an SEBS-coated micro-buckle surface ($R(\varepsilon)$ is the resistance of fibre at the stretched state and $L_{max}$ is the length of fibre at the fully stretched state). **b** Amplitude of square wave voltage generated by functional generator ($f = 20$ Hz) versus time recorded at initial, and 1000% strain application for pristine (up panel), and SEBS-coated (down panel) supercoiled fibres. **c** Quality factor (percent strain divided by percent resistance change) versus strain for spandex@SEBS/CNT supercoiled fibre ($Q = 238.8$, green star), and comparison with previously reported fibre electrodes: (a) a sandwich-structured CNT/rubber fibre ($Q = 54$)[28], (b) a buckled CNT/SEBS fibre ($Q = 65.1$)[10], (c) a coiled CNT/nylon fibre ($Q = 0.1$)[3], (d) a buckled, coiled CNT/rubber fibre ($Q = 1.8$)[17], and (e) a CNT/elastomer polymer fibre ($Q = 1.6$)[29]. Inset shows retention performance versus average voltage amplitude for pristine (open circles), and SEBS-coated (solid circles) supercoiled fibres. Schematic illustrations and images of the measurement setup for **d** electrocardiogram (ECG), **e** audio, and **f** video signal transmissions using supercoil fibres (all supercoiled fibres were mounted on Vernier calipers for measurements). Resulting **g** ECG (consisting of P wave, QRS wave, and T wave), **h** audio, and **i** video signals transmitted by commercial cables (reference, black line), and supercoil fibres of nonstretched (red line), and 1000% stretched (green line) are compared

Moreover, the signals did not distort even at 1000% strain applications as shown in Fig. 4h, i, respectively. Successful demonstrations of audio and video signal transmission during dynamic supercoil fibre stretching/releasing (up to 1000%) were demonstrated in Supplementary Movie 1 and 2, respectively.

**Electrochemical performance of fibrous supercapacitor with high stretchability.** A remarkable advantage of the presented supercoiled fibres is that they can serve as effective electrodes for a highly deformable electrochemical energy storage system by simply loading electrochemically active materials. To demonstrate superelastic fibre supercapacitors, we used the pseudocapacitive $MnO_2$, which is abundant and environmentally friendly with a high theoretical specific capacitance, as active material.

Pseudocapacitive, supercoil fibre electrodes were prepared by drop-casting a $MnO_2$ particle-dispersed solution (20 mg mL$^{-1}$) directly onto the surface of as-fabricated spandex@CNT supercoil fibres, resulting in $MnO_2$/CNT composite electrodes (Supplementary Fig. 7). Symmetrical, electrochemical capacitors were prepared using two parallel configured spandex@$MnO_2$/CNT supercoiled fibres. To complete the fabrication of the solid-state supercapacitor, an aqueous poly(vinyl alcohol) (PVA)/LiCl gel electrolyte was used for coating. By controlling the number of drop-casting, the resulting $MnO_2$ loading level could be further increased up to 17.7 wt%. cyclic voltammetry (CV) curves measured at 50 mV s$^{-1}$ for solid-state supercoil supercapacitors with various $MnO_2$ loading levels are compared and related specific capacitance is plotted in Fig. 5a, b, respectively. The box-like rectangular CV curves indicated no observable Faradic redox

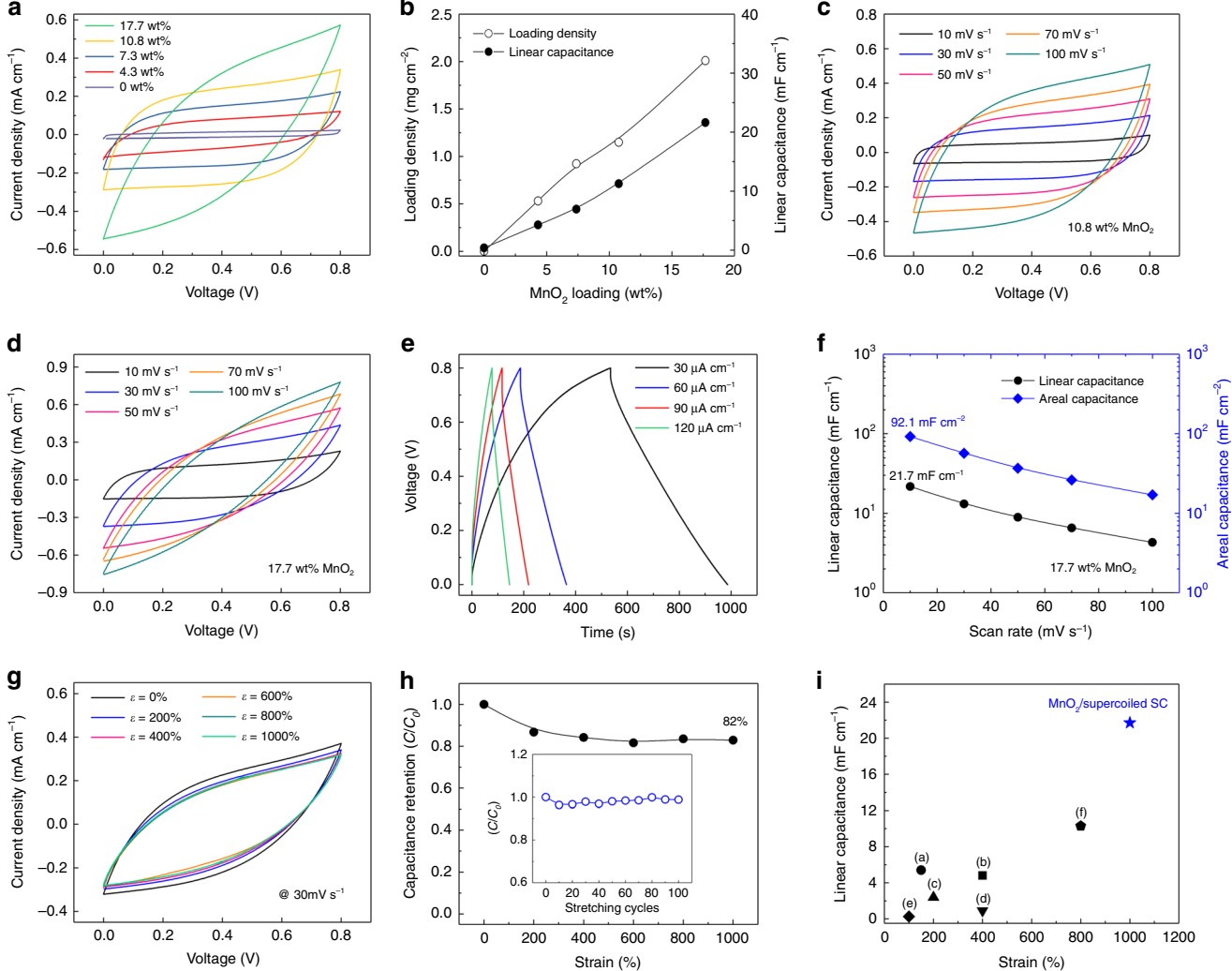

**Fig. 5** Electrochemical performance of solid gel electrolyte coated supercapacitor. **a** Cyclic voltammetry (CV) curves at 50 mV s⁻¹ for various MnO₂ wt%. **b** Calculated MnO₂ areal loading density, and linear capacitances versus MnO₂ loading wt%. CV curves (from 10 to 100 mV s⁻¹) of supercoil fibres with **c** 10.8 wt%, and **d** 17.7 wt% MnO₂ loadings. **e** Galvanostatic charge/discharge curves (current density from 25 to 100 μA cm⁻¹) of supercapacitor (SC) 17.7 wt% MnO₂ loading. **f** Specific linear (blue diamond) and areal (black circle) capacitance values versus scan rate. **g** CV curves measured for the initial ($\varepsilon$ = 0%) and statically stretched states of the supercoil supercapacitor. **h** Capacitance retention performance versus tensile strain (inset shows capacitance retention versus cyclically applied stretching test). **i** Linear capacitances versus maximum strains of various fibre supercapacitors. The maximum linear capacitance ($C_L$ = 21.7 mF cm⁻¹) and stretchability ($\varepsilon$ = 1000%) for the present supercoil supercapacitor exceed (a) a coiled MnO₂/carbon nanotube (CNT)/nylon fibre (5.4 mF cm⁻¹, 150%)[3], (b) a coiled MnO₂/CNT/rubber fibre (4.8 mF cm⁻¹, 400%)[17], (c) a MnO₂/CNT/rectangular rubber sandwich fibre (2.38 mF cm⁻¹, 200%)[28], (d) a polyaniline (PANI)/CNT/elastomer fibre (0.9 mF cm⁻¹, 400%)[29], (e) a spandex@CNT-based wire SC (0.26 mF cm⁻¹, 100%)[35], and (f) a styrene-ethylene-butylene-styrene (SEBS) wounded by CNT/graphene/PANI fibres (10.3 mF cm⁻¹, 800%)[25]

reaction for our supercoil supercapacitor, and the curves were consistent with energy storage by an electrochemical double-layer capacitor (EDLC) from CNT and pseudocapacitance from MnO₂[3]. Although the bare spandex@CNT supercoil can be an EDLC-based capacitor, the specific capacitances can be dramatically increased so that about 56-times larger CV curves were obtained after 17.7 wt% MnO₂ decoration. The CV curves measured from 10 to 100 mV s⁻¹ scan rates for 10.8 and 17.7 wt% MnO₂ loadings are presented in Fig. 5c, d, respectively. It is observed that the CV curves from higher MnO₂ loaded supercapacitor are dented at higher scan rates. The degradation of rate-capability at high MnO₂ loading originates from its low electrical conductivity (~ 10⁻⁵ to 10⁻⁶ S cm⁻¹), which significantly limits charge transport[32]. Although the nanoscopically thin MnO₂ film reduces the solid-state ion diffusion lengths and provides high specific capacitance per MnO₂ weight, because of

extremely low loading mass, the overall energy and power densities (including all components) of MnO₂-based devices have been low[33]. The galvanostatic charge/discharge curves for the 17.7 wt% MnO₂-decorated, supercoil supercapacitor are also presented in Fig. 5e. The maximum linear and areal specific capacitances of the MnO₂-decorated, supercoil supercapacitor versus the scan rate were calculated from the CV curves to be 21.7 mF cm⁻¹ and 92.1 mF cm⁻² (based on single-electrode capacitances), respectively; these values were achieved at 10 mV s⁻¹ scan rate (Fig. 5f). Investigation of the effects of mechanical deformations on the electrochemical performances of a MnO₂-decorated supercoil supercapacitor was performed using a PVA/LiCl gel electrolyte. As the LiCl exhibits excellent dehumidification effect[34], the PVA/LiCl contains the water molecule, and can maintain quasi-solid state with high viscosity even when it is exposed to air. The CV curves for the supercoil supercapacitor

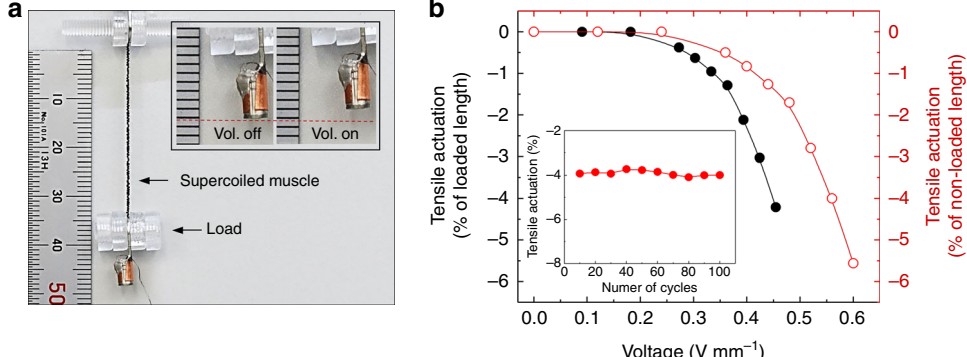

**Fig. 6** Performance of supercoiled fibre-based artificial muscle. **a** Photograph showing supercoiled fibre artificial muscle under 175 kPa isobaric load. Inset images show weight, hanged by supercoiled fibre, before and after electrical voltage application. **b** Tensile strokes normalised by loaded and nonloaded fibre length versus applied voltage for supercoiled fibre (inset shows tensile actuation versus cyclically applied voltage)

| Table 1 Comparison of stretchability and functionality of various coil structured fibres | | | | |
| --- | --- | --- | --- | --- |
| **Structure (ref. no.)** | **Electrode materials** | **Fabrication** | **Stretchability [%]** | **Functionality** |
| Supercoil (this work) | Spandex@CNT fibres | Over-twisting | ~ 800–1500<br>1000<br>4.2 | Transmission line, supercapacitor, artificial muscle |
| Coil, ply[20] | CNT fibres | Twisting | 24 | Artificial muscle |
| Coil[3] | Nylon@CNT fibres | Twisting | 150 | Supercapacitor |
| Coil[21] | Nylon fibres | Twisting | 49 | Artificial muscle |
| Coil[14] | Graphene oxide fibres | Twisting | 76 | Heating element |
| Coil[22] | PVA/graphene fibres | Twisting | 400 | High toughness |
| Helical coil[23] | PEDOT-S:PSS fibres | Fibre winding | 400 | Supercapacitor |
| Helical coil[24] | CNT/graphene/MnO$_2$ fibres | Fibre winding | 850 | Supercapacitor |
| Helical coil[25] | SEBS@CNT/graphene/PANI fibres | Fibre winding | 800 | Supercapacitor |

*CNT* carbon nanotube, *PVA* poly(vinyl alcohol)

were measured under application of static tensile strains (Fig. 5g). Negligible changes were observed in the CV curves (100 mV s$^{-1}$ scan rate in a two-electrode system) when the strain applied varied from 0 to 1000% in the tensile direction. The capacitance retention performance versus tensile strain is characterised in Fig. 5h. About 82% of the capacitance of the supercoil supercapacitor was retained at 1000% strain application. Moreover, the supercoil supercapacitor showed highly stable capacitance retention (98.8%) against a cyclically applied 1000% tensile strain (inset in Fig. 5h). Optical image of Supplementary Fig. 8 shows PVA/LiCl gel coated, two parallel supercoil spandex@MnO$_2$/CNT fibres at 1000% strain application. This indicates that the transparent gel electrolyte is well surrounded between the fibre electrodes without degrading the electrodes' stretchability. The linear specific capacitance of the supercoiled supercapacitor (SC) versus maximum stretchability is plotted in Fig. 5i, and compared with previously reported core–shell structured or elastomeric substrate-based stretchable fibre or yarn-based supercapacitors. Both linear capacitance ($C_L = 21.7$ mF cm$^{-1}$) and maximum stretchability ($\varepsilon = 1000\%$) for the present supercoil supercapacitor are higher than that of previously reported fibre or yarn-based supercapacitors with similar microscale diameters: (a) MnO$_2$/CNT@coiled nylon fibre SC ($C_L = 5.4$ mF cm$^{-1}$, $\varepsilon = 150\%$)[3], (b) buckled MnO$_2$/CNT@coiled rubber fibre SC ($C_L = 4.8$ mF cm$^{-1}$, $\varepsilon = 400\%$)[17], (c) buckled MnO$_2$/CNT@sandwich structured rubber fibre SC ($C_L = 2.38$ mF cm$^{-1}$, $\varepsilon = 200\%$)[28], (d) a polyaniline (PANI)/CNT@elastomer polymer fibre SC ($C_L = 0.9$ mF cm$^{-1}$, $\varepsilon = 400\%$)[29], (e) spandex@MnO$_2$/CNT and spandex@CNT fibres for wire-shaped asymmetric SC ($C_L = 0.26$ mF cm$^{-1}$, $\varepsilon = 100\%$)[35], and (f) spring structure CNT/graphene/PANI fibres ($C_L = 10.3$ mF cm$^{-1}$,

$\varepsilon = 800\%$)[25]. Not only linear capacitance, but also areal capacitances of the supercoil supercapacitor ($C_A = 92.1$ mF cm$^{-2}$) were also comparable to previously reported supercapacitors with lower stretchability values (Supplementary Table 2).

**Performance of supercoiled fibre-based artificial muscle.** The supercoiled spandex@CNT fibre can also function as electro-thermally driven artificial muscles to provide large contractile stroke under heavy loads. The actuation performance for a supercoiled fibre artificial muscle was measured during the application of a voltage for Joule heating (Fig. 6a). The CNT sheath acts as a heating element to induce volume change of core spandex fibre during voltage application. The tensile stroke of a polymer muscle can be amplified by inserting a large amount of twisting[21]. The supercoil fibre-based muscles delivered maximum contractile tensile actuation of 4.2% based on a loaded fibre length (or 5.5% actuation based on nonloaded fibre length) when 0.45 V mm$^{-1}$ was applied under a load of 175 kPa (Fig. 6b). The stress is obtained by normalising to the nonactuated, nontwisted fibre cross-sectional area. The specific work capacity of the supercoil muscle during contraction was 0.17 kJ kg$^{-1}$. The supercoiled muscle also showed stable actuation performance during repeated voltage application (inset of Fig. 6b).

**Discussion**

In summary, we developed a highly twisted supercoil structure of superelastic conducting fibres with various potential applications. Table 1 compares the fabrication method, stretchability, and functionality of the supercoil to prior-art various coil structured

fibres for clarifying the novelty of this work. We showed that the fibres can be used as a cable or transmission line that are extendable up to 11 times their initial length without significant resistance change for applications such as ECG or other audio and video signal transmissions. Additionally, the introduction of electrochemically active materials enables the assigning of pseudocapacitive energy storage functionality into the supercoiled fibres, resulting in superelastic supercapacitor fibres. When electrical energy was applied, the supercoiled fibres showed electrothermally operated tensile actuation originating from the core fibre volume expansion, which was amplified by the insertion of twist. Other possible applications for such highly elastic fibres include the morphing of structures in space, robotic arms or exoskeletons capable of extreme reach, and interconnects for highly elastic electronic circuits[1].

## Methods

**Supercoiled fibre fabrication**. Multiwalled carbon nanotube (MWNT) sheets were mechanically drawn from a carbon nanotube (CNT) forest with 750-μm height (NTAD 10, PDSI Corporation, Korea) and were wrapped on a 200 μm-diameter commercially available spandex fibre (Hyosung, Korea). The spandex fibre was stretched (400%) during the CNT wrapping. After wrapping the CNT on the spandex fibre, the spandex@CNT fibre was highly twisted for full supercoiling. For supercoiled fibre passivation, the SEBS particles were dissolved in cyclohexane (Sigma-Aldrich, USA) and the solution was spray-coated on the supercoiled fibre while it was fully stretched. For supercapacitor electrodes fabrication, MnO$_2$ nanoparticles (Sigma-Aldrich, USA) were dispersed in alcohol (20 mg mL$^{-1}$) using sonication, and the solution was drop-cast onto the surface of the spandex@CNT fibre before supercoiling. PVA/LiCl gel was used as electrolytes for electrochemical performance characterisation. More specifically, the gel electrolyte was prepared by mixing 3 g PVA (Sigma-Aldrich, USA) and 6 g LiCl (Sigma-Aldrich, USA) in 30 mL deionized water and heating it at 90 °C until it became transparent.

**Characterisation**. MWNT sheet wrapping was performed using homemade fibre twisting machines. Images of fibre morphology were obtained with an SEM (S-4600, Hitachi, Japan) and an optical camera (D750, Nikon, Japan). For electrical resistance measurements for fibres during the statically applied tensile strain, the fibres were mounted on digital Vernier calipers (500 series, Mitutoyo, Japan), with both fibre ends mechanically fixed by a bolt–nut pair, and electrically connected to multimeter probes (15 +, Fluke) for resistance measurements. A function generator (AFG1062, Tektronix) and an ECG measurement system (MP36, Biopac) were used to investigate the electrical signal transmission performance. An electrochemical analyser (Vertex EIS, Ivium) was used for electrochemical performance characterisation.

**Calculation of the electrochemical and actuation performances**. The capacitance of the two-electrode system was calculated from CV curves. From $C = I/(dV/dt)$, where $I$ is average discharge current from the CV curve and the $dV/dt$ is the scan rate, the single-electrode specific areal capacitance was calculated from the following Eq. 1:

$$\text{Areal capacitance} \left( \text{F cm}^{-2} \right) = 2C/A_{\text{surface}} \tag{1}$$

where $A_{\text{surface}}$ is the total external surface area of a single spandex@MnO$_2$/CNT fibre electrode. The total fibre length was used for linear capacitance calculation. For actuation performance characterisations, the fibre length changes were measured during voltage application using a voltage source (E3633A, Agilent). The tensile stroke of the supercoiled fibre was calculated using Eq. 2:

$$\text{Contraction tensile stroke}(\%) = (L_{\text{final}} - L_{\text{initial}})/L_{\text{initial}} \times 100 \tag{2}$$

where $L_{\text{initial}}$ is the initial supercoiled fibre length under isobaric load and $L_{\text{final}}$ is the final fibre length under isobaric load after voltage application. The specific work capacity per mass was calculated using Eq. 3:

$$\text{Work capacity} \left( \text{kJ kg}^{-1} \right) = m_{\text{load}} gh/m_{\text{fibre}} \tag{3}$$

where $m_{\text{load}}$ is isobaric load, $g$ is gravitational acceleration, $h$ is the change in fibre height, and $m_{\text{fibre}}$ is the mass of the supercoiled fibre.

## Data availability
The data that support the findings of this study are available from the corresponding author on reasonable request.

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

## Acknowledgements

This work was supported by the Basic Science Research Programme (NRF-2017R1A6A3A04004987) and Global Research and Development Center Program (NRF-2018K1A4A3A01064272) through the National Research Foundation of Korea funded by the Ministry of Education and Ministry of Science and ICT. This work was also supported by the DGIST R&D Programme (18-NT-02) of the Ministry of Science, ICT, and Future Planning.

## Author contributions

W.S., S.C., J.M.L. and C.C. conceived the idea and designed the experiments; D.W.L. and Y.K. contributed mechanical/electrochemical characterisation; S.M.J. fabricated material for experiments; W.S., S.C., S.K.L. and C.C. wrote the paper; Y.L., J.P., H.J., Y.J.K. and D.S. performed the signal transmission experiments; All authors discussed the results and commented on the manuscript.

## Additional information

**Competing interests:** The authors declare no competing interests.

