## [Peer Review File · Nature Communications]

Reviewers' Comments:

Reviewer #1:

Remarks to the Author:

The author reported a kind of superelastic fiber based on a supercoil structure. The fiber shows outstanding strain to 1300% due to the coil structure. However, the coil-strategy is not new (Adv. Mater. 2015, 27, 4982–4988, 10.1016/j.joule.2018.06.003). Moreover, the title is not appropriate. It looks more like an "over twisted yarn" other than "DNA", based on the SEM images provided in Figure 1f. Besides, the demonstrated applications seems ordinary without showing the advantage of 1300% superelastic. I suggest to make the novelty of this work clearer by better design.

Following are some specific comments.

Major issue:

1. The resulting supercoiled fiber is fabricated by inserting a giant twist into spandex core fibers wrapped in a carbon nanotube sheath, which the authors call "CNT@spandex fiber". This definition of the supercoiled fiber should be "spandex@CNT fiber" because carbon nanotubes are the sheath of the fiber.
2. The author could show more in-depth study of the special structure.
3. How to realize the large-production of the supercoil fibers? What is the length of the fiber achieved in this article? As far as I know, the over twisted process is not stable. The author should convince the readers of the possibility of large-production.
4. The mechanical properties should be added. And "as effective and universal substrates for various wearable applications", the mechanical properties should meet the standard for textile fabrication.
5. The application of fibers showed in Figure 3e can be replaced by other materials which have the same effect and lower cost. Such high strain (1000%) is not needed in ECG wires. The author should explain the necessity of superelastic fiber in ECG monitoring. Otherwise, it is meaningless.
6. Some references should be added between 400%-1000% in Figure 4f. Recently published super-elastic supercapacitors should be cited and discussed to give the readers a full picture.
7. Pseudocapacitive supercoil fiber electrodes are prepared by drop-casting an MnO₂ particle-dispersed solution onto the surface of as-fabricated supercoil fibers. What is the percentage of MnO₂ in resulting electrodes? How will the percentage of each component affect the properties of resulting electrodes?
8. Although the resulting supercoiled supercapacitor has excellent stretchability that can sustain tensile strain up to 1000%, the PVA/LiCl gel electrolyte can not sustain tensile strain of 1000%. How did the authors prepare the supercapacitor with strain up to 1000%? Please explain.

Reviewer #2:

Remarks to the Author:

This paper introduces a method to fabricate stretchable fibers with a coiled configuration and tested their mechanical and electrical performance. There are strong interests to develop fabric-based stretchable structures. The concept in this paper is interesting but this reviewer has the following concerns:

- (1) The title "NDA-inspired" is somewhat misleading. The stretchable fiber is just coil-shaped, or more specifically a fiber with coil shapes in two length scales. They are not really double helix structures.
- (2) The formation of the first buckled structure, the main innovation of this paper, was not clearly discussed. To form a buckled structure, the CNT sheath layers must have certain mechanical properties, such as mechanical modulus, and thickness. These factors all affect the buckling geometry and thus affects the stretchability. They are needed to be thoroughly discussed.
- (3) It is also unclear how the geometry of the second level coil (e.g., figure 1h) affects the performance. Again, thorough characterizations AND discussions are needed.

Reviewer #3:

Remarks to the Author:

The authors reported that superelastic and electrically conducting fibers fabricated by inserting twist into spandex core fibers wrapped in a carbon nanotube sheath. They have demonstrated performance of developed fibers as supercapacitors and artificial muscles. Since fabrication of superelastic and electrically conducting has already reported (J. Foroughi, Knitted Carbon-Nanotube-Sheath/Spandex-Core Elastomeric Yarns for Artificial Muscles and Strain Sensing, ACS Nano. 2016, 10, 9129-9135., Z. Lu, J., Superelastic Hybrid CNT/Graphene Fibers for Wearable Energy Storage, Advanced Energy Materials 2017. , Liu,Z. F. et al. Hierarchically buckled sheath-core fibers for superelastic electronics, sensors, and muscles, Science 2015)

It is expected to see some novelty in the current reported. However, there is no significant improvement on performance of developed fibers including electrochemical actuation, energy storage or fabrications process. In addition one significant issue is how to prevent fabricated fibers from uncoiling. Since coiled sample has to be under tension and there is explanation how to keep sample without release of twist and permanently set. Thermally annealing process was used for the coiled nylon or other thermoplastic polymer however Spandex (polyether-polyurea copolymer) has a very low Tg (below room temperature) and impossible to annealing method.

I believe that the manuscript should be considered for other journal and it is not suitable for Nature Comm. Due to insufficient novelty and innovations

Response to Reviewer Comments

We appreciate the comments of the reviewers and the suggestions of the editor. Our responses to these comments are listed below, and the manuscript and supplemental materials have been accordingly revised. To help address the comments of the reviewers, we performed additional experiments and added pertinent new results. This includes (1) demonstration of large production possibility, more advanced applications, and free-standing supercoil fibres, (2) structural analysis, and characterization of structural effects on electrical/mechanical properties, (3) MnO₂ weight percent measurement, electrochemical performance optimization, and the capacitance retention of PVA-LiCl gel electrolyte coated 1,000% stretchable supercapacitor. Main revision contents are summarized in table as shown below. Moreover, we revised the key word ‘fiber’ into ‘fibre’ for NPG publication, and recalculate the length-normalized twisting number based on not the final length but the initial length of the fibre before twisting insertion in order to fairly compare the twisting numbers with other references.

Reviewer No.	Revisions or new results based on reviewer suggestions	Revised or newly added figure
# 1	- Novelty enhancement by comparing present work with previously reported various coil structured fibres	- Fig. S1, Table 1
	- Structural analysis on microscale buckles	- Fig. 2 (a), (b)
	- Structural analysis on macroscale supercoils	- Fig. 2 (c) ~ (g)
	- Demonstration of possibility of large production	- Fig. S4, 5
	- Stress-strain curves and possibility of textile fabrication	- Fig. 3 (d), Fig. 1 (k)
	- Demonstration of advanced applications	- Fig. 4 (e), (f), (h), (i), Movie S1, S2
	- MnO ₂ wt% calculation and performance optimization	- Fig. 5 (a) ~ (d)
	- Demonstration of 1,000% stretchable PVA-LiCl gel-coated SC	- Fig. 5 (g), (h), Fig. S8
# 2	- Structural analysis on microscale buckles	- Fig. 2 (a), (b)
	- Structural analysis on macroscale supercoils	- Fig. 2 (c) ~ (g)
# 3	- Novelty enhancement by comparing present work with previously reported various coil structured fibres	- Fig. S1, Table 1
	- Demonstration of free-standing supercoil fibre	- Fig. 1 (h), (i)

Reviewer #1 (Remarks to the Author)

The author reported a kind of superelastic fiber based on a supercoil structure. The fiber shows outstanding strain to 1300% due to the coil structure. However, the coil-strategy is not new (Adv. Mater. 2015, 27, 4982–4988, 10.1016/j.joule.2018.06.003). Moreover, the title is not appropriate. It looks more like an “over twisted yarn” other than “DNA”, based on the SEM images provided in Figure 1f. Besides, the demonstrated applications seems ordinary without showing the advantage of 1300% superelstic. I suggest to make the novelty of this work clearer by better design. Following are some specific comments.

Response: Thank you for your kind comment about interest in our work. Although the coil-strategy is not new as a *Reviewer #1* commented, presented supercoil is apparently a new structure that has not been reported yet. To help the readers clearly understand the structural difference of supercoil with others, we newly added a schematic illustrations (**Figure S1**) showing structural differences of various coil structured fibres.

Figure S1. Schematic illustrations showing (a) supercoiled, (b) coiled, and (c) helical coiled fibres

Table 1 also compares fabrication method, stretchability, and functionality of the supercoil and prior-art various coil structured fibres (including the references *Reviewer #1* referred [9, 10]) to clarify the novelty of this work. For examples, the first-coil fibres are widely reported for various stretchable fibrous applications such as supercapacitors, actuators, and high toughness [1-5], and can be fabricated by moderate twisting insertions. Due to the simple structure, however, the first-coiled fibres normally reported less than 400% stretchability. For helical-coil fibres, meanwhile, the main difference with other coils is that they are fabricated by not just twisting but by winding active fibres onto the surface of non-active core substrate fibres [6-8]. The helical-coil fibres are reported to be stretchable up to 850% when the core substrates have appropriate elasticity. However, they can suffer from low specific performances when

normalized by whole system dimension including thick and bulk core substrate. This is because the as-used core substrate does not contribute to the fibre functionalities (energy storage or actuation) but just provides a mechanical stretchability, which might especially lead to low specific capacitances, energy, or power densities. Presented supercoils are unique structure that is fabricated by only highly over-twist insertion and due to the high degree of structural compaction, they exhibited superelasticity (~ 1,500%) without significant loss in electrical property. We newly added **Figure S1, Table 1** and added this information in **page 3, line 8** and **page 15, line 17** in the revised manuscript.

Table 1. Comparison of stretchability and functionality of present and prior-art various coil structured fibres or textiles.

Structure (Ref. No.)	Electrode Materials	Fabrication	Stretchability [%]	Functionality
Supercoil (this work)	Spandex@CNT fibres	Over-twisting	800 ~ 1,500 1,000 4.2	Transmission line, supercapacitor, artificial muscle
Coil, ply (1)	CNT fibres	Twisting	24	Artificial muscle
Coil (2)	Nylon@CNT fibres	Twisting	150	Supercapacitor
Coil (3)	Nylon fibres	Twisting	49	Artificial muscle
Coil (4)	Graphene oxide fibres	Twisting	76	Heating element
Coil (5)	PVA/graphene fibres	Twisting	400	High toughness
Helical coil (6)	PEDOT-S:PSS fibres	Fibre winding	400	Supercapacitor
Helical coil (7)	CNT/graphene/MnO ₂ fibres	Fibre winding	850	Supercapacitor
Helical coil (8)	SEBS@CNT/graphene/PANI fibres	Fibre winding	800	Supercapacitor
2D coil (9)	SEBS embedded Li/Cu coils	Wrapping in spiral pattern	60	Li-ion battery
Knitted textile (10)	Spandex@CNT fibres	Knitting	100	Strain sensor

A *Reviewer #1* also pointed out that the supercoil looks more like an “over twisted yarn” other than “DNA”, based on the SEM images provided in Figure 1f. According to the comments, we revised our title from “DNA-Inspired Supercoils for Superelastic Fibres” into “Highly Twisted

Supercoils for Superelastic Fibres”. In addition, we removed related key words (DNA, inspiration), and scheme (figure 1a showing DNA structure in previous version manuscript). Finally, we added new demonstrations of supercoil fibre applications, which will be discussed later, in order to showing advantage of present work better.

Major issue:

(1) The resulting supercoiled fiber is fabricated by inserting a giant twist into spandex core fibers wrapped in a carbon nanotube sheath, which the authors call "CNT@spandex fiber". This definition of the supercoiled fiber should be "spandex@CNT fiber" because carbon nanotubes are the sheath of the fiber.

Response: We revised all “CNT@spandex fiber” into “spandex@CNT fibre” in revised manuscript, and have made appropriate changes elsewhere.

(2) The author could show more in-depth study of the special structure.

Response: According to the valuable comments, we performed additional experiments to study the structural effects, and characterization is included in **Figure 2** which is newly added in revised version manuscript. Our unique structures which contribute to high fibre stretchability can be categorized into two parts: one is microscopic scale buckles, and the other is macroscopic scale supercoils. The morphological width of CNT micro-buckles is investigated against pre-strain (**Figure 2a**) applied to a bare, noncoiled spandex fibre before CNT wrapping, and CNT loading layers (**Figure 2b**). Upon observations, the width of the buckles is roughly, inversely proportional to the degree of pre-strain application (CNT loading level is fixed as five layers in this case). The average width of the CNT buckles formed from relaxation of 100% pre-strained fibre was about 38 μm and it significantly decreased to 16.3 μm by 400% pre-strain relaxation. The effect of CNT loading level on buckles formation, where pre-strain is fixed at 400%, showed slight increase of buckle width from 14.6 to 20 μm as the number of CNT wrapping layer increased from 1 to 9. The fibre contraction force to recover initial length after pre-strain relaxation is inversely proportional to the fibre length ratio of before to after

relaxation ($\Delta L/L_0$) as shown in fig. 2b. The larger contraction force by either larger pre-strain application or lower CNT loading level results in narrower buckle width.

Figure 2. Measured width of microscopic CNT buckles of non-coiled, relaxed spandex@CNT fibre versus (a) pre-strain applied to the pristine spandex fibre before CNT wrapping, and (b) number of CNT layer for wrapping.

Meanwhile, a main experimental parameter to control various morphological conditions of the supercoil (**Figure 2c**) is the total twisting number inserted to complete the full supercoil formation. Linear coil density (number of coils per unit length) and supercoil index (D'/d) are plotted versus twisting number insertion as shown in **Figure 2d**. Here, linear coil density can be defined as N/L , where N is total number of coils for a given fibre length (L), and can be calculated by dividing a fibre length by coil width (L/D). Therefore, the linear coil density per centimeter fibre length is $N/L = (L/D)/L = 1/D$. While the coil density linearly increased as the twisting number increases, the coil index did not show any observable or meaningful tendencies. In addition, the aspect ratio (L/D') of the supercoil fibres, in case of supercoil fabrication starting from 7 cm-long initial spandex fibre, varied from 12.6 to 42.2 with increasing twisting number as shown **Figure 2e**. In other words, higher twisting number causes tougher squeezing of the fibre during supercoiling, leading to thinner and longer supercoiled fibres. In this reason, both fibre stretchability and change in resistance during the fibre stretching decreased accordingly by higher number of twisting insertion as shown **Figure 2e**. The maximum stretchability of the supercoil fibre fabricated with 4,165 turns/m is about 1,500% and steadily drops to 600% for the fibre fabricated with 10,496 turns/m. Optical and SEM images for supercoils with low (4,165 turns/m, **Figure 2f**) and high (10,496 turns/m, **Figure 2g**) total twisting numbers were then compared. The microscopic surface from the higher twisting number inserted supercoil fibre was less buckled and largely tightened, which

might be caused by the additional tensile strain generated by inserting more twists. We newly added **Figure 2** and the characterization on buckle, and supercoil structures in **page 6, line 18** in the revised manuscript.

Figure 2. (c) Scheme showing various morphological parameters of supercoil fibre. (d) Linear coil density ($1/D$) and supercoil index (D'/d) versus total twisting number insertion for fully supercoiling. (e) Fibre aspect ratio (L/D'), maximum stretchability, and resistance change during fibre maximum stretching versus total twisting number insertion for fully supercoiling. Optical images showing supercoil fibres fabricated from (f) low twisting (4,165 turn/m), and (g) high twisting (10,496 turn/m) (both scale bars = 300 μm). Insets show SEM images of magnified coil surfaces (both scale bars = 100 μm).

(3) How to realize the large-production of the supercoil fibers? What is the length of the fiber achieved in this article? As far as I know, the over twisted process is not stable. The author should convince the readers of the possibility of large-production.

Response: The final length of the most supercoil fibres presented in this work was mostly less than 2 cm for lab-scale fabrication. The fabrication processes were started from stretching 7 cm-long initial spandex fibre up to 400% in tensile direction (35 cm in length), and wrapping CNTs onto the surface of the stretched spandex fibres. Over-twisting process is a widely used and reliable strategy to fabricate the coiled yarns or fibres as previously reported [3, 5]. To demonstrate the possibility of the large-production of supercoil fibres, we newly fabricated and demonstrated the 60 cm-long supercoil fibre as shown image below (two ends of the fibre were tethered using bolts and nuts). In fabrication detail, both ends of bare spandex fibre with 3 m-long were fixed to the twisting motor tips and 400% pre-strain was applied (15 m in length). After appropriate CNT wrapping the spandex@CNT fibre was relaxed from pre-strain and the relaxed fiber length was 3.5 m. Total 7,000 turns/m of twisting was inserted for supercoiling and 60 cm-long supercoil fibre was successfully fabricated. Formation of supercoil was stable that the supercoil morphology observed by optical microscope was largely uniform as shown images below (**Figure S4**).

Figure S4. Photograph (upper) and magnified optical images (lower) for four different spots of 60 cm-long supercoil fibre (scale bar = 500 μm). Two ends of the fibre are tethered by bolt/nut pairs to prevent the untwisting.

Electrical property of the 60 cm-long supercoil fibre was also investigated by measuring fibre resistances versus fibre length as shown below (**Figure S5**). The fibre exhibited resistance-linearity property with 100 Ω/cm slope value. The inset photograph shows 60 cm-long supercoil fibre wound in 1 cm-diameter glass tube. Therefore, with high length of the fibre, we confirmed that the morphological and electrical properties are uniform and the over-

twisting method is revisable. We added this data in **Supplementary figure S4, S5** and **page 6, line 12** in the revised manuscript.

Figure S5. Measured electrical resistance versus length of 60 cm-long supercoil fibre. Inset shows photograph of 60 cm-long supercoil fibre wounded on 1 cm-diameter glass tube.

(4) The mechanical properties should be added. And “as effective and universal substrates for various wearable applications”, the mechanical properties should meet the standard for textile fabrication.

Response: Mechanical property of the supercoil fibre was characterized by measuring engineering stress-strain curves as shown in **Figure 3d** in revised manuscript. The supercoil fibre fabricated from 5,000 turns/m twisting insertion exhibited about 80 MPa ultimate tensile strength and break at 1,500% in tensile strain. The inset shows comparison of first and 100th stress loading-unloading curves for 1,000% strain. The curve shows reversible and stable mechanical properties under repeated strain application. We added this mechanical property characterization data and description in **Figure 3d** and **page 9, line 6**, respectively, in the revised manuscript. Possibility of textile fabrication using the supercoil fibres is demonstrated in **Figure 1j**, and **k**. The supercoiled fibres were mechanically strong enough to be woven into a commercial textile that six-supercoil fibres were successfully sewn into mock rib-structured textile (**Figure 1j**). Moreover, the supercoil fibres can be also assembled into the textile themselves which comprising twenty-seven spandex@CNT fibres (**Figure 1k**).

Figure 3. (d) Engineering stress-strain curves for 150µm-diameter supercoil fibre. Inset shows stress loading-unloading curves comparison before (black line) and after (red line) 100th 1000% strain application.

Figure 1. Optic images for **j**) six-woven supercoiled spandex@CNT fibres into a commercial mock rib-structured textile, and **k**) 20 mm-long, 7 mm-wide supercoil textile consisting of twenty-seven spandex@CNT fibres.

(5) The application of fibers showed in Figure 3e can be replaced by other materials which have the same effect and lower cost. Such high strain (1000%) is not needed in ECG wires. The author should explain the necessity of superelastic fiber in ECG monitoring. Otherwise, it is meaningless.

Response: We appreciate your important comments. In the manuscript, we could successfully fabricated passivation layer (SEBS) over-coated spandex@CNT supercoil fibres in order to dramatically reduce the resistance change during the fibre stretching. As a result, the fibre exhibited a 4.2% resistance increase for a stretch of 1,000% strain application, which can be quantified as an impressively higher quality factor (238.8) compared with previous stretchable fibrous conductors. One of possible, and remarkable application for present work is the signal transmission cable that are extendable up to 16 times their initial length, which can be applied

for morphing structures in space, robotic arms or exoskeletons capable of extreme reach, or interconnects for highly elastic electronic circuits [11]. Moreover, a few groups reported stretchable signal transmission lines with strain-insensitive resistance that remains nearly constant even when these lines are subjected to large strains. Such properties are desired for transmitting analog voltage or current signals, for example, which may contain vital information from sensors attached to or implanted in a human body. Combined with high signal-to-noise ratios (SNRs), these stretchable conductors are likely to be very promising in health care and wound-monitoring systems, or related applications [12, 13].

Based on the *Reviewr #1*'s suggestion, we performed additional experiments of better designed applications showing the advantage of 1,000% superelasticity. **Figure 4d, e and f** show photograph images for experimental setups for various signal transmissions. **Figure 4g, h and i** compare the signals' amplitude transmitted by reference commercial BNC cable and present supercoil fibres (before and after 1,000% strain applications). The audio and video signals were split into the supercoil fibre and reference line simultaneously for comparison. Real-time signal transmissions and comparisons under the dynamic strain application were characterized for audio (see **Movie S1**) and video (see **Movie S2**) signals. In general, relevant frequencies range from 20 Hz to 20 kHz for acoustic electronics devices and are on the order of a few MHz for composite video signals; these frequencies are much higher than those considered in ECG measurements [14]. In principle, transmission of video signals requires much higher accuracy in terms of the signal period (or frequency) and amplitude because the information on brightness, color, and synchronization of a signal is delivered at the same time [14]. Specifically, various significant signals such as horizontal synchronization, color reference burst, active pixel region, and front and back porch should be respectively transmitted without delay and distortion [14]. The transmitted signals by supercoil fibres were almost identical with that of reference cables without distorting the original information in terms of its resistance and frequency. Moreover, the signals did not distorted even at 1000% strain applications as shown in **Figure 4h, and i**. We added this audio and video signal transmission data and description in **Figure 4e, f, h, i and Movie S1, S2, and page 11, line 4**, in the revised manuscript.

Figure 4. Application demonstrations of supercoil fibres as effectively superelastic transmission lines. **(f)** Photograph showing experimental set-up for audio signal transmission and **(g)** received voltage amplitude comparisons between commercial cable (reference) and SEBS/supercoiled fibre before and after 1000% strain application. **(h)** Photograph showing video signal transmission experiment, and **(i)** the resulting voltage amplitude comparison.

(6) Some references should be added between 400% - 1000% in Figure 4f. Recently published super-elastic supercapacitors should be cited and discussed to give the readers a full picture.

Response: We added three more recently reported literatures about stretchable supercapacitors with high stretchability (400~850%) [6-8] and compared the specific capacitances with present work as shown **Supplementary table S2** and **Figure 5i**. It should be noted that, by increasing the MnO₂ loading level during the revision process, the maximum measured linear and areal capacitances were increased up to 21.7 mF/cm, and 92.1 mF/cm² for the presently investigated superelastic supercapacitors consisting of symmetrical spandex@MnO₂(17.7wt%)/CNT based supercoil fibres coated by PVA/LiCl gel electrolyte.

Supplementary table S2. Comparison of specific capacitances and stretchability of present work and prior-art various stretchable fibres supercapacitors.

Electrode Materials (Ref. No.)	C_L [mF/cm]	C_A [mF/cm ²]	Stretchability [%]
Spandex@MnO ₂ /CNT supercoiled fibres (this work)	21.7	92.1	1,000
MnO ₂ /CNT@coiled nylon fibres (2)	5.4	40.9	150
Buckled MnO ₂ /CNT@coiled rubber fibres (15)	4.8	22.8	400
Buckled MnO ₂ /CNT@sandwich structured rubber fibres (16)	2.4	11.9	200
PANI/CNT@elastomeric polymer fibres (17)	0.9	50.1	400
MnO ₂ /CNT@spandex, CNT@spandex asymmetric fibres (18)	0.26	27.9	100
SEBS@CNT/graphene/PANI helical-coil fibres (8)	10.3	273.7	800
PEDOT-S:PSS helical-coil fibre (6)		93.1	400
CNT/graphene/MnO ₂ helical-coil fibres (7)		14.02	850

The linear capacitance versus strain for fibrous stretchable supercapacitors was plotted and compared as shown in **Figure 5i**. The present work has advantageous structure to store extremely high strain in tensile direction based on compact and well-organized supercoil structure. Therefore, the presented linear capacitance value as long as stretchability far exceeds equivalent values from previously reported stretchable fiber based supercapacitors: (a) MnO₂/CNT@coiled nylon fibre SC ($C_L = 5.4$ mF/cm, $\varepsilon = 150\%$) [2], (b) buckled MnO₂/CNT@coiled rubber fibre SC ($C_L = 4.8$ mF/cm, $\varepsilon = 400\%$) [15], (c) buckled MnO₂/CNT@sandwich structured rubber fibre SC ($C_L = 2.38$ mF/cm, $\varepsilon = 200\%$) [16], (d) PANI/CNT@elastomer polymer fibre SC ($C_L = 0.9$ mF/cm, $\varepsilon = 400\%$) [17], (e) MnO₂/spandex@CNT and spandex@CNT fibres for wire-shaped asymmetric SC ($C_L = 0.26$ mF/cm, $\varepsilon = 100\%$) [18], and (f) spring structure CNT/graphene/PANI fibres ($C_L = 10.3$ mF/cm, $\varepsilon = 800\%$) [8]. We added this performance comparison in **Supplementary table S2, Figure 5i** in the revised manuscript.

Figure 5. i) Linear capacitances versus maximum strains of various fibre supercapacitors. The maximum linear capacitance ($C_L = 21.7$ mF/cm) and stretchability ($\epsilon = 1000\%$) for the present supercoil supercapacitor significantly exceed [a] a coiled MnO₂/CNT/nylon fibre supercapacitor (5.4 mF/cm, 150%)², [b] a coiled MnO₂/CNT/rubber fibre (4.8 mF/cm, 400%)¹⁵, [c] a MnO₂/CNT/rectangular rubber sandwich fibre (2.38 mF/cm, 200%)¹⁶, [d] a PANI/CNT/elastomer fibre (0.9 mF/cm, 400%)¹⁷, [e] a spandex@CNT-based wire SC (0.26 mF/cm, 100%)¹⁸, and [f] a SEBS wounded by CNT/graphene/PANI fibres (10.3 mF/cm, 800%)⁸. All measurements were performed from solid-state supercapacitors comprising two symmetric spandex@MnO₂/CNT supercoil fibres coated by PVA-LiCl electrolyte.

(7) Pseudocapacitive supercoil fiber electrodes are prepared by drop-casting an MnO₂ particle-dispersed solution onto the surface of as-fabricated supercoil fibers. What is the percentage of MnO₂ in resulting electrodes? How will the percentage of each component affect the properties of resulting electrodes?

Response: Based on your comment, we experimentally measured weight percent (wt %) of drop-casted MnO₂ particles using microbalance (XM 1000P, DWS inc.), and performed additional experiments to characterize the MnO₂ loading level effect. For spandex@MnO₂/CNT supercoil electrode in previous manuscript, it is measured that about 4.3 wt% MnO₂ (based on total supercoil fibre weight including spandex core, CNT, and MnO₂) was loaded. By controlling the number of drop casting, the resulting MnO₂ loading level could be further increased up to 17.7 wt%. The CV curves measured at 50 mV/sec for solid-state supercoil supercapacitors with various MnO₂ loading levels are compared and related specific capacitance is plotted as shown in **Figure 5a** and **b**, respectively. The linear, and areal specific capacitances are significantly increased from 4.3 mF/cm, 22.4 mF/cm² to 21.7 mF/cm, 92.1

mF/cm², respectively, by higher MnO₂ loading (17.7 wt%). Despite the capacitance increase, however, the rate-capability was apparently decreased. The CV curves measured from 10 to 100 mV/sec scan rates for 10.8 and 17.7 wt% MnO₂ loadings are presented in **figure 5c and d**, respectively. It is observed that the CV curves from higher MnO₂ loaded supercoil supercapacitor gets dented at higher scan rate. This degradation of rate-capability at high MnO₂ loading is originated from low electrical conductivity ($\sim 10^{-5}$ to 10^{-6} S/cm), which significantly limits charge transport [19]. Although a nanoscopically thin MnO₂ film reduces the solid-state ion diffusion lengths and provides high specific capacitance per MnO₂ weight, because of extremely low loading mass, the overall energy and power densities (including all components) of MnO₂-based devices have been low [20]. We added this MnO₂ wt% characterization and performance optimization **Figure 5a, b, c, and d** presented in **page 13, line 1, 8** in the revised manuscript.

Figure 5. Electrochemical performances of PVA-LiCl gel electrolyte coated supercoil fibre supercapacitors. **(a)** CV curves with various MnO₂ wt %. **(b)** Calculated linear capacitances versus MnO₂ wt% loading. CV curves measured from 10 to 100 mV/s for **(c)** 10.8 wt%, and **(d)** 17.7 wt% of MnO₂ loadings.

(8) Although the resulting supercoiled supercapacitor has excellent stretchability that can sustain tensile strain up to 1000%, the PVA/LiCl gel electrolyte can not sustain tensile strain of 1000%. How did the authors prepare the supercapacitor with strain up to 1000%? Please explain.

Response: In the manuscript, PVA/LiCl was used as an electrolyte for electrochemical performance characterization. As the LiCl has excellent dehumidification effect [21], the PVA/LiCl contains the water molecule, and can maintain quasi-solid state with high viscosity. In this reason, some previously reported highly stretchable supercapacitors used the PVA/LiCl gel as a stretchable electrolyte [2, 15]. Optical image of **Figure S8** shows PVA/LiCl gel coated, two parallel supercoil spandex@MnO₂/CNT fibres at 1000% strain application. It is shown that the transparent gel electrolyte is well surrounded between the fibre electrodes without degrading the electrodes' stretchability.

Figure S8. Optical image of supercoil fibre based solid-state supercapacitor at 1000% tensile strain application, which comprises of two parallel, symmetric spandex@MnO₂/CNT supercoil fibres and PVA-LiCl gel electrolyte coating (scale bar = 300µm).

Therefore, the presented electrochemical data measured during strain application in revised manuscript (**Figure 5g, h**) were replaced from electrochemical results of PVA/LiCl coated supercoil supercapacitor. We added this mechanical performance characterization data and description in **Supplementary figure S8**, and **Figure 5g, h**, and **page 13, line 23**, and **page 14, line 8** in the revised manuscript.

Figure 5. (g) CV curves measured for the initial ($\epsilon = 0\%$) and statically stretched states of the supercoil supercapacitor. (h) Capacitance retention performance versus tensile strain (inset shows capacitance retention versus cyclically applied stretching test).

Reviewer #2 (Remarks to the Author):

This paper introduces a method to fabricate stretchable fibers with a coiled configuration and tested their mechanical and electrical performance. There are strong interests to develop fabric-based stretchable structures. The concept in this paper is interesting but this reviewer has the following concerns:

(1) The title "DNA-inspired" is somewhat misleading. The stretchable fiber is just coil-shaped, or more specifically a fiber with coil shapes in two length scales. They are not really double helix structures.

Response: We appreciate your very kind comment about the importance of our work. According to the comments, we removed related key words ("DNA" and "inspiration"), and scheme showing DNA structure in the revised manuscript to prevent the authors to misunderstand our work. Moreover, we revised our title from "DNA-Inspired Supercoils for Superelastic Fibres" into "Highly Twisted Supercoils for Superelastic Fibres".

(2) The formation of the first buckled structure, the main innovation of this paper, was not clearly discussed. To form a buckled structure, the CNT sheath layers must have certain mechanical properties, such as mechanical modulus, and thickness. These factors all affect the buckling geometry and thus affects the stretchability. They are needed to be thoroughly discussed.

Response: According to the valuable comments, we performed additional experiments to study the structural effects, and characterization is included in **Figure 2** which is newly added in revised version manuscript. The morphological width of CNT micro-buckles is investigated against pre-strain (**Figure 2a**) applied to a bare, noncoiled spandex fibre before CNT wrapping, and CNT loading layers (**Figure 2b**). Upon observations, the width of the buckles is roughly, inversely proportional to the degree of pre-strain application (CNT loading level is fixed as five layers in this case). The average width of the CNT buckles formed from relaxation of 100% pre-strained fibre was about 38 μm and it significantly decreased to 16.3 μm by 400% pre-strain relaxation. The effect of CNT loading level on buckles formation, where pre-strain is fixed at 400%, showed slight increase of buckle width from 14.6 to 20 μm as the number of CNT wrapping layer increased from 1 to 9. The fibre contraction force to recover initial length

after pre-strain relaxation is inversely proportional to the fibre length ratio of before to after relaxation ($\Delta L/L_0$) as shown in fig. 2b. The larger contraction force by either larger pre-strain application or lower CNT loading level results in narrower buckle width.

Figure 2. Measured width of microscopic CNT buckles of non-coiled, relaxed spandex@CNT fibre versus (a) applied pre-strain before CNT wrapping, and (b) number of CNT layer wrapping.

(3) It is also unclear how the geometry of the second level coil (e.g., figure 1h) affects the performance. Again, thorough characterizations AND discussions are needed.

Meanwhile, a main experimental parameter to control various morphological conditions of the supercoil (**Figure 2c**) is the total twisting number inserted to complete the full supercoil formation. Linear coil density (number of coils per unit length) and supercoil index (D'/d) are plotted versus twisting number insertion as shown in **Figure 2d**. Here, linear coil density can be defined as N/L , where N is total number of coils for a given fibre length (L), and can be calculated by dividing a fibre length by coil width (L/D). Therefore, the linear coil density per centimeter fibre length is $N/L = (L/D)/L = 1/D$. While the coil density linearly increased as the twisting number increases, the coil index did not show any observable or meaningful tendencies. In addition, the aspect ratio (L/D') of the supercoil fibres, in case of supercoil fabrication starting from 7 cm-long initial spandex fibre, varied from 12.6 to 42.2 with increasing twisting number as shown **Figure 2e**. In other words, higher twisting number causes tougher squeezing of the fibre during supercoiling, leading to thinner and longer supercoiled fibres. In this reason, both fibre stretchability and change in resistance during the fibre

stretching decreased accordingly by higher number of twisting insertion as shown **Figure 2e**. The maximum stretchability of the supercoil fibre fabricated with 4,165 turns/m is about 1,500% and steadily drops to 600% for the fibre fabricated with 10,496 turns/m. Optical and SEM images for supercoils with low (4,165 turns/m, **Figure 2f**) and high (10,496 turns/m, **Figure 2g**) total twisting numbers were then compared. The microscopic surface from the higher twisting number inserted supercoil fibre was less buckled and largely tightened, which might be caused by the additional tensile strain generated by inserting more twists. We newly added **Figure 2** and the characterization on buckle, and supercoil structures in **page 6, line 19** in the revised manuscript.

Figure 2. (c) Scheme showing various morphological parameters of supercoil fibre. (d) Linear coil density ($1/D$) and supercoil index (D'/d) versus total twisting number insertion for fully supercoiling. (e) Fibre aspect ratio (L/D'), maximum stretchability, and resistance change during fibre maximum stretching versus total twisting number insertion for fully supercoiling. Optical images showing supercoil fibres fabricated from (f) low twisting (4,165 turn/m), and (g) high twisting (10,496 turn/m) (both scale bars = 300 μm). Insets show SEM images of magnified coil surfaces (both scale bars = 100 μm).

Reviewer #3 (Remarks to the Author):

The authors reported that superelastic and electrically conducting fibers fabricated by inserting twist into spandex core fibers wrapped in a carbon nanotube sheath. They have demonstrated performance of developed fibers as supercapacitors and artificial muscles. Since fabrication of superelastic and electrically conducting has already reported (J. Foroughi, Knitted Carbon-Nanotube-Sheath/Spandex-Core Elastomeric Yarns for Artificial Muscles and Strain Sensing, ACS Nano. 2016, 10, 9129-9135., Z. Lu, J., Superelastic Hybrid CNT/Graphene Fibers for Wearable Energy Storage, Advanced Energy Materials 2017. , Liu, Z. F. et al. Hierarchically buckled sheath-core fibers for superelastic electronics, sensors, and muscles, Science 2015) It is expected to see some novelty in the current reported. However, there is no significant improvement on performance of developed fibers including electrochemical actuation, energy storage or fabrications process.

Response: Thank you for your valuable comments about our work. Although the coil-strategy is not new, present supercoil is apparently a new structure that has not been reported yet. It seems that the two references *referee #3* referred [Superelastic Hybrid CNT/Graphene Fibers for Wearable Energy Storage, Advanced Energy Materials 2017. , Liu,Z. F. et al. Hierarchically buckled sheath-core fibers for superelastic electronics, sensors, and muscles, Science 2015] can be classified into first-coil or helical-coil structures, which are different from present supercoil. Moreover, we believe that the other reference [J. Foroughi, Knitted Carbon-Nanotube-Sheath/Spandex-Core Elastomeric Yarns for Artificial Muscles and Strain Sensing, ACS Nano. 2016, 10, 9129-9135.] is also different from our work in term their work and performances are based on not the fibre but textile structure. To help the readers clearly understand the structural difference of present supercoil with others, we newly added a schematic illustrations (**Figure S1**) showing structural differences of various coil structured fibres.

Figure S1. Schematic illustrations showing (a) supercoiled, (b) coiled, and (c) helical coiled fibres

Table 1 also compares fabrication method, stretchability, and functionality of the supercoil and prior-art various coil structured fibres to clarify the novelty of this work. For examples, the first-coil fibres are widely reported for various stretchable fibrous applications such as supercapacitors, actuators, and high toughness [1-5], and can be fabricated by moderate twisting insertions. Due to the simple structure, however, the first-coiled fibres normally reported less than 400% stretchability. For helical-coil fibres, meanwhile, the main difference with other coils is that they are fabricated by not just twisting but by winding active fibres onto the surface of non-active core substrate fibres [6-8]. The helical-coil fibres are reported to be stretchable up to 850% when the core substrates have appropriate elasticity. However, they can suffer from low specific performances when normalized by whole system dimension including thick and bulk core substrate. This is because the as-used core substrate does not contribute to the fibre functionalities (energy storage or actuation) but just provides a mechanical stretchability, which might especially lead to low specific capacitances, energy, or power densities. Presented supercoils are unique structure that is fabricated by only highly over-twist insertion and due to the high degree of structural compaction, they exhibited superelasticity (~1,500%) without significant loss in electrical property. We newly added **Figure S1, Table 1** and added this information in **page 3, line 8** and **page 15, line 17** in the revised manuscript.

Table 1. Comparison of stretchability and functionality of present and prior-art various coil structured fibres or textiles.

Structure (Ref. No.)	Electrode Materials	Fabrication	Stretchability [%]	Functionality
Supercoil (this work)	Spandex@CNT fibres	Over-twisting	800 ~ 1,500 1,000 4.2	Transmission line, supercapacitor, artificial muscle
Coil, ply (1)	CNT fibres	Twisting	24	Artificial muscle
Coil (2)	Nylon@CNT fibres	Twisting	150	Supercapacitor
Coil (3)	Nylon fibres	Twisting	49	Artificial muscle
Coil (4)	Graphene oxide fibres	Twisting	76	Heating element
Coil (5)	PVA/graphene fibres	Twisting	400	High toughness
Helical coil (6)	PEDOT-S:PSS fibres	Fibre winding	400	Supercapacitor
Helical coil (7)	CNT/graphene/MnO ₂ fibres	Fibre winding	850	Supercapacitor
Helical coil (8)	SEBS@CNT/graphene/PANI fibres	Fibre winding	800	Supercapacitor
2D coil (9)	SEBS embedded Li/Cu coils	Wrapping in spiral pattern	60	Li-ion battery
Knitted textile (10)	Spandex@CNT fibres	Knitting	100	Strain sensor

In addition one significant issue is how to prevent fabricated fibers from uncoiling. Since coiled sample has to be under tension and there is explanation how to keep sample without release of twist and permanently set. Thermally annealing process was used for the coiled nylon or other thermoplastic polymer however Spandex (polyether-polyurea copolymer) has a very low T_g (below room temperature) and impossible to annealing method. I believe that the manuscript should be considered for other journal and it is not suitable for Nature Comm. Due to insufficient novelty and innovations

Response: Torque-balance locking can be an effective way to make the supercoil fiber free-standing state, which is previously reported for CNT coil yarns [22]. Photograph for double-helix structured supercoil fibers made by self-inter locking does not need tethering as shown image below (**Figure 1h**, and **i**). This type of internally torque-balanced structure eliminates

the need for external torsional tethering. Plying a single, fully supercoiled fibre in the opposite direction to the fibre's internal twist creates a structure in which the chiralities of fibre twist and plying are opposite. This plying was accomplished by folding the supercoiled fibre itself, prohibiting relative rotation of fibre ends. Another possible strategy to prohibit the untwisting is to upscale the supercoil fibres into neat textile form. The supercoil fibres were mechanically strong enough to be woven into commercial mock-rib structured textile (**Figure 1j**). In this case, the woven supercoil fibres were mechanically fixed by adjacent fibres of neat structured textile, therefore, no any observable untwisting was generated while they were woven. The supercoil fibres could be also fully assembled into textile structure by themselves (**Figure 1k**). We added this demonstration of free-standing supercoiled fibre, and textile assembly in **Figure 1h, i, j, and k**, and **page 6, line 1**, respectively, in the revised manuscript.

Figure 1. Photograph for h) free-standing state double-helix structured supercoil fibre, and optic images for i) its magnification (scale bar = 1 mm), j) six-woven supercoiled spandex@CNT fibres into a commercial mock rib-structured textile, and k) 20 mm-long, 7 mm-wide supercoil textile consisting of twenty-seven spandex@CNT fibres.

References

- [1] Chen, P. *et al.* Electromechanical Actuator Ribbons Driven by Electrically Conducting Spring-Like Fibers. *Adv. Mater.* **27**, 4982-4988 (2015).
- [2] Choi, C. *et al.* Stretchable, Weavable Coiled Carbon Nanotube/MnO₂/Polymer Fiber Solid-State Supercapacitors. *Sci. Rep.* **5**, 9387 (2015).
- [3] Haines, C. S. *et al.* Artificial Muscles from Fishing Line and Sewing Thread. *Science* **343**, 868 (2014).
- [4] Cruz-Silva, R. *et al.* Super-stretchable Graphene Oxide Macroscopic Fibers with Outstanding Knotability Fabricated by Dry Film Scrolling. *ACS Nano* **8**, 5959-5967 (2014)
- [5] Zhang, J. *et al.* Multiscale deformations lead to high toughness and circularly polarized emission in helical nacre-like fibres. *Nat. Commun.* **7**, 10701 (2016).
- [6] Wang, z. *et al.* All-in-one fiber for stretchable fiber-shaped tandem supercapacitors. *Nano Energy* **45**, 210-219 (2018).
- [7] Wang, H. *et al.* Superelastic wire-shaped supercapacitor sustaining 850% tensile strain based on carbon nanotube@graphene fiber. *Nano Research* **11**, 2347-2356 (2018).
- [8] Lu, Z. *et al.* Superelastic Hybrid CNT/Graphene Fibers for Wearable Energy Storage. *Adv. Energy Mater.* **8**, 1702047 (2018).
- [9] Liu, K. *et al.* Stretchable Lithium Metal Anode with Improved Mechanical and Electrochemical Cycling Stability. *Joule* **2**, 1-9 (2018).
- [10] Foroughi, J. *et al.* Knitted Carbon-Nanotube-Sheath/Spandex- Core Elastomeric Yarns for Artificial Muscles and Strain Sensing. *ACS Nano* **10**, 9129-9135 (2016).
- [11] Liu, Z. F. *et al.* Hierarchically buckled sheath-core fibers for superelastic electronics, sensors, and muscles. *Science* **349**, 400-404 (2015).
- [12] Lewis, K. M. *et al.* Sensible Analysis of the 12-lead ECG. *Delmar: Tohmson Learning* (2000).
- [13] Inoue, S. *et al.* Video microscopy. *Springer Science & Business Media New York* (1986).
- [14] Lee, Y. *et al.* Ultrastretchable Analog/Digital Signal Transmission Line with Carbon Nanotube Sheets. *ACS Appl. Mater. Interfaces* **9**, 26286-26292 (2017).

- [15] Choi, C. *et al.* Microscopically Buckled and Macroscopically Coiled Fibers for Ultra-Stretchable Supercapacitors. *Adv. Energy Mater* **7**, 1602021 (2017).
- [16] Choi, C. *et al.* Twistable and Stretchable Sandwich Structured Fiber for Wearable Sensors and Supercapacitors. *Nano Lett.* **16**, 7677-7684 (2016).
- [17] Zhang, Z. *et al.* Superelastic Supercapacitors with High Performances during Stretching. *Adv. Mater.* **27**, 356-362 (2015).
- [18] Xu, P. *et al.* Stretchable Wire-Shaped Asymmetric Supercapacitors Based on Pristine and MnO₂ Coated Carbon Nanotube Fibers. *ACS Nano* **9**, 6088-6096 (2015).
- [19] Xu, C. *et al.* Recent progress on manganese dioxide based supercapacitors. *J. Mater. Res.* **25**, 1421-1432 (2010).
- [20] Xu, C., Kang, F., Li, B. & Du, H. Recent progress on manganese dioxide based supercapacitors. *J. Mater. Res.* **25**, 1421-1432 (2010)
- [21] Zhang, L.-Z. *et al.* Synthesis and characterization of a PVA/LiCl blend membrane for air dehumidification. *Journal of Membrane Science* **308**, 198-206 (2008).
- [22] Lee, J. A. *et al.* Electrochemically Powered, Energy-Conserving Carbon Nanotube Artificial Muscles. *Adv. Mater.* **29**, 1700870 (2017).

Reviewers' Comments:

Reviewer #1:

Remarks to the Author:

This manuscript was revised and the reply to the comments are clear and reasonable. I would like to recommend acceptance of this paper for publication.

Minor issues:

1) In Figure 3a, "supercoiled CNT/spandex fibre" should be revised as "supercoiled spandex@CNT fibre".

2) All usage of abbreviations should be defined at the first occurrence in the text, such as MWNT in the Methods.

Reviewer #2:

Remarks to the Author:

The authors added some new results to address my original comments 2 and 3. From the perspective of completion, this manuscript is acceptable. However, the novelty of this work does not meet the high-standard of Nature Communications.

Reviewer #3:

Remarks to the Author:

I would be happy with the current version and it could be considered for publication in this journal

REVIEWERS' COMMENTS:

Reviewer #1 (Remarks to the Author):

This manuscript was revised and the reply to the comments are clear and reasonable. I would like to recommend acceptance of this paper for publication.

Minor issues:

- 1) In Figure 3a, "supercoiled CNT/spandex fibre" should be revised as "supercoiled spandex@CNT fibre".
- 2) All usage of abbreviations should be defined at the first occurrence in the text, such as MWNT in the Methods.

Response: Thank you for your kind response to our work. We revised "supercoiled CNT/spandex fibre" into "supercoiled spandex@CNT fibre" in Figure 3a, and added definitions of all abbreviations at their first occurrence in revised manuscript according to your valuable comments.

Reviewer #2 (Remarks to the Author):

The authors added some new results to address my original comments 2 and 3. From the perspective of completion, this manuscript is acceptable. However, the novelty of this work does not meet the high-standard of Nature Communications.

Response: Thank you for your kind response to our work.

Reviewer #3 (Remarks to the Author):

I would be happy with the current version and it could be considered for publication in this journal

Response: Thank you for your kind response to our work.